# A parasite DNA binding protein with potential to influence disease susceptibility acts as an analogue of mammalian HMGA transcription factors

Zeeshan Durrani[1], Jane Kinnaird[2], Chew Weng Cheng[3], Francis Brühlmann[4],
Paul Capewell[2], Andrew Jackson[1], Stephen Larcombe[5], Philipp Olias[4,6], William Weir[2],
Brian Shiels[2]*

**1** Institute of Infection, Veterinary and Ecological Sciences, University of Liverpool, Liverpool, United Kingdom, **2** School of Biodiversity, One Health and Veterinary Medicine, College of Medical, Veterinary and Life Sciences, University of Glasgow, Glasgow, United Kingdom, **3** Leeds Institute of Cardiovascular and Metabolic Medicine, School of Medicine, University of Leeds, Leeds, United Kingdom, **4** Institute of Animal Pathology, Vetsuisse Faculty, University of Bern, Bern, Switzerland, **5** School of Biological Sciences, Ashworth Laboratories, University of Edinburgh, Edinburgh, United Kingdom, **6** Institute of Veterinary Pathology, Justus Liebig University, Giessen, Germany

* brian.shiels@glasgow.ac.uk

**Data Availability Statement:** All primary data detailed in this study is available in the paper and

## Abstract

Intracellular pathogens construct their environmental niche, and influence disease susceptibility, by deploying factors that manipulate infected host cell gene expression. *Theileria annulata* is an important tick-borne parasite of cattle that causes tropical theileriosis. Excellent candidates for modulating host cell gene expression are DNA binding proteins bearing AT-hook motifs encoded within the *TashAT* gene cluster of the parasite genome. In this study, *TashAT2* was transfected into bovine BoMac cells to generate three expressing and three non-expressing (opposite orientation) cell lines. RNA-Seq was conducted and differentially expressed (DE) genes identified. The resulting dataset was compared with genes differentially expressed between infected cells and non-infected cells, and DE genes between infected cell lines from susceptible Holstein vs tolerant Sahiwal cattle. Over 800 bovine genes displayed differential expression associated with TashAT2, 209 of which were also modulated by parasite infection. Network analysis showed enrichment of DE genes in pathways associated with cellular adhesion, oncogenesis and developmental regulation by mammalian AT-hook bearing high mobility group A (HMGA) proteins. Overlap of TashAT2 DE genes with Sahiwal vs Holstein DE genes revealed that a significant number of shared genes were associated with disease susceptibility. Altered protein levels encoded by one of these genes (*GULP1*) was strongly linked to expression of *TashAT2* in BoMac cells and was demonstrated to be higher in infected Holstein leucocytes compared to Sahiwal. We conclude that TashAT2 operates as an HMGA analogue to differentially mould the epigenome of the infected cell and influence disease susceptibility.

its Supporting Information files. The RNA sequence datasets generated and analyzed during the current study are publicly available from the BioStudies repository (https://www.ebi.ac.uk/arrayexpress/experiments/E-MTAB-9680).

**Funding:** BS was awarded a grant from the Biotechnology and Biological Sciences Research Council (BBSRC): https://www.ukri.org/councils/bbsrc/ Grant BB/L004739/1 The funder had no role in study design, data collection and analysis, decision to publish, or preparation of the manuscript.

**Competing interests:** The authors have declared that no competing interests exist

## Introduction

In order to establish infection, intracellular pathogens must construct an environmental niche within host cells. The infected cell niche in turn has potential to promote pathology and drive the evolution of host-parasite interaction and susceptibility to disease [1]. Numerous examples exist of pathogens constructing a favourable niche [2, 3]. One mechanism that has capacity to generate widespread changes in the intracellular environment is the export of parasite molecules that perturb the regulation of host cell gene expression [4–8].

The tick-borne apicomplexan parasite *Theileria annulata* is one of several *Theileria* species that cause economically important disease in ruminants throughout tropical and sub-tropical regions of the Old World [9]. *Theileria annulata* together with *T. parva* and *T. lestoquardi* are known as 'transforming' *Theileria* species because, following sporozoite invasion of leukocytes and differentiation to the macroschizont, the parasite generates a 'cancer-like' transformed host cell [10]. Economic losses due to tropical theileriosis are significant and constrain introduction of productive cattle breeds into countries where the parasite is endemic [10–13].

Transformation of bovine leukocytes by *Theileria* parasites involves activation of host cell transcription factors (TF) that promote proliferation and prevent apoptosis but have the potential to induce a pro-inflammatory response [14]. Thus, macroschizonts of *T. annulata* constitutively activate NF-κB and prevent apoptosis [15, 16]. However, while some NF-κB target genes show a predicted elevated expression others, including those that function in the pro-inflammatory response are negatively regulated [17]. Similar control is exerted over the interferon (IFN) pathway; infected cells generate type I IFNs [18], but are refractory to IFN stimulation and moderate expression of IFN stimulated genes (ISGs) [19]. These results suggest the parasite exerts tight control over the activation outcome of the infected cell via extensive parasite-dependent tuning of host cell gene expression levels [17, 20].

*Theileria* factors that rewire the host transcriptome have not been conclusively defined. Comparative genomics has identified several gene families that are unique to transforming *Theileria* species [21]. Of these, the *TashAT/TashHN* gene cluster currently provides the best candidates for tuning of host cell gene expression. The majority of genes in the cluster encode proteins expressed by the macroschizont stage that are predicted to be transported to the infected leukocyte nucleus [14], with several detected in this location [22–24]. TashAT cluster proteins have been shown to bind DNA [24] and some (TashAT1-3 and SuAT1-3) contain AT-hook domains. These domains were initially identified in the mammalian HMGA1 protein [25]. HMGA1 and other members of the HMGA gene family act as architectural transcription factors which influence the epigenetic landscape that determines accessibility of transcription factors to chromatin. This function, in turn, regulates patterns of gene expression in inflammation and cancer [26]. A consensus DNA motif recognised by TashAT2 has been defined and as expected, based on the AT-hook domain, consists of two runs of AT rich sequence interspersed with a short region richer in G and C nucleotides [27]. Moreover, in our recent study, variability in number and pattern of the TashAT2 consensus motif was associated with gene loci that display differential expression in parasite infected leukocytes derived from cattle breeds either tolerant or susceptible to tropical theileriosis [28].

Evidence that *TashAT* genes could modulate the host cell has been obtained by expression of TashAT cluster genes in the macrophage-derived BoMac cell line. Transfected BoMacs displayed altered cell morphology [19, 24] and modulated expression of the IFN-stimulated genes (ISG), UBP43 (alias USP18) and ISG15 [19]. However, the full extent to which TashAT2 can modulate host gene expression was not fully determined, due to a lack of sensitivity of the methodology used to identify differentially expressed transcripts.

To robustly determine the full capability of TashAT2 to modulate bovine gene expression, we have generated six independently transfected, non-infected bovine macrophage (BoMac) cell lines that are either stably expressing the TashAT2 transgene or are transfected with a non-expressing construct (reverse orientation). RNA-Seq was performed and differentially expressed genes for *TashAT2* expressing cells identified. Comparative analysis was then conducted to test whether TashAT2 target genes are also a) modulated by parasite infection, b) associated with susceptibility to disease and c) similar to known target genes of mammalian HMGA proteins.

## Material and methods

### Cell lines and culture

The SV40 transformed cell line (BoMac) derived from bovine macrophages [29], was maintained in culture at 37˚C in RPMI-1640 (Gibco, ThermoFisher Scientific, UK) supplemented with 10% heat-inactivated foetal calf serum (Sigma Aldrich, UK), 8 μg/mL streptomycin/8 units/mL penicillin (Gibco), 0.6 μg/mL amphotericin B (Gibco) and 0.05% $NaHCO_3$ (Gibco). Transfected cell lines were selected and maintained with geneticin (G418) at 500 μg/mL (InvivoGen, Europe). The infected bovine cell lines derived from Sahiwal (S3) and Holstein (H3) cattle were generated and cultured as described [28, 43], and represent the Hissar (India) isolate of *T. annulata*.

### Generation of constructs for transfection

The *TashAT2* gene was originally cloned from a *Theileria annulata* genomic DNA library [22]. Generation of V5-tagged *TashAT2* constructs was as described in Oura *et al.*, 2005 [19]. Briefly, the cloned *TashAT2* open reading frame with a V5 epitope tag was cloned into the mammalian cell expression vector pCAGGsIRESBGeo (pCAGGs, kindly provided by Nicholas Gilbert, University of Edinburgh, described and referenced in Oura *et al.* [19]. This vector contains an internal ribosome entry sequence (IRES) and a chicken beta actin promotor. As V5-*TashAT2* was blunt-end ligated into the pCAGGs vector, plasmids containing either forward or reverse orientations of the insert were obtained. Orientation was determined by single and double digestion with *BamHI*, *EcoRI* and *HindIII*. Following geneticin selection, lines generated expressing the V5 tag were termed pKP38 (pCAGGsIRESGeo: *TashAT2*) and pKP37 (pCAGGsIRESGeo: *TashAT2*rev, i.e. *TashAT2* in reverse orientation).

### Transfection of constructs

BoMac cells were grown until sub-confluent ($\sim 2.8 \times 10^6$ cells per 75 $cm^2$ flask). Prior to transfection by electroporation using a BioRad Gene Pulsar II and conditions recommended by the manufacturer, cells were trypsinised, washed and then resuspended in RPMI containing 10 mM Glucose (Sigma Aldrich, UK)/0.1 mM dithiothreitol (DTT, Sigma Aldrich, UK) without antibiotics. For each transfection, $2.0 \times 10^6$ cells were used and the amount of DNA based on plasmid size. Thus, for pCMVEGFP-C1, 2 μg; pKP38, 6 μg; pKP37, 6 μg. DNA was pre-incubated with cells in the cuvette for 5 min then electroporated at 960 μF, 200 V. Following 5 min recovery, the cells were transferred to 15 mL of medium pre-equilibrated with $CO_2$. Geneticin was added to 500 μg/mL after 36 h and established colonies were obtained after 17 days of drug selection. All transfected cell cultures were fed by replacement with fresh medium. When reasonably vigorous growth was evident, frozen stocks were generated using standard methodology. The selected replicate cell lines used in this study were each derived from three independent transfections carried out with the constructs: pKP37 (i.e. *TashAT2* gene in reverse

orientation) or pKP38, (*TashAT2* in forward orientation). The plasmid pCMV EGFP-C1 was used as a transfection efficiency control at 24 h post-transfection.

## Immunofluorescence and immunoblotting

Expression of the V5 tag and TashAT2 in each of the independently selected pKP38 cell lines was determined by immunofluorescence (IFAT) on cells deposited on cytospin slides, as described previously [19]. Absence of tag/transgene protein expression was also confirmed by IFAT in the three negative control pKP37 lines. Following trypsinisation and washing in phosphate buffered saline (PBS), cells were cytospun, fixed in 3.7% paraformaldehyde (Sigma Aldrich, UK) pH 7.4 for 30 min at 4°C, followed by methanol for 10 min at -20°C. Slides were then washed X3 in PBS. An anti-V5 mouse monoclonal antibody (Sigma Aldrich, UK) and a rabbit polyspecific Ab raised against TashAT2 fusion protein [22] were used at 1/500 dilution. Detection of primary antibodies was performed with anti-mouse or anti-rabbit Alexa 488 secondary antibody (Invitrogen, ThermoFisher Scientific, UK) at 1/200 dilution. Images with matched exposures were captured using an Olympus BX60 microscope, SPOT camera and SPOT™ Advanced image software Version Mac: 4.6.1.26 (Diagnostic Instruments). Preparation of BoMac, infected cell line extracts and immunoblotting was performed as described [24, 28]. Immunoblots were probed with rabbit polyspecific antisera against GULP1 (PA5-54240, ThermoFisher Scientific, UK) diluted 1/600 and TashAT2 [22] diluted 1/1000.

## RNA preparation

RNA was prepared from parallel log-phase cultures representing each of the six transfected cell lines. RNA was isolated using TRIzol reagent (ThermoFisher Scientific, UK) with a further purification using RNeasy columns (Qiagen), including an on-column DNase digestion step, following the supplier's protocol. RNA quality and quantity were then assessed using the Nanodrop ND-1000 spectrophotometer (ThermoFisher Scientific, UK) and by gel electrophoresis. Detailed methodology of RNA extraction, DNase treatment, clean-up and quantification of RNA was described previously [17, 20].

## RNA sequencing (RNA-Seq)

RNA-Seq was performed at the Centre for Genomic Research, University of Liverpool. A minimum of 1200 ng DNase treated total RNA was submitted per sample for RNA-Seq library preparation and sequencing using the Illumina HiSeq 2500 platform. A single lane of the instrument was used to sequence all six samples. The RNA-Seq generated $\sim$54 million paired-end 150 bp reads for each sample. The raw reads were processed and assessed using CASAVA version 1.8.2 (Illumina). The base calling and de-multiplexing of indexed reads resulted in the creation of sequence data files in FASTQ format. PCR duplicates and low quality reads were trimmed from the raw reads using Cutadapt version 1.2.1 [30]. The parameter "-O 3" was used to trim off the 3′ end of any reads that matched the adapter sequence over at least 3 bp. Initial quality control (QC) to evaluate the reads was performed using FastQC [31]. The UMD3.1 version of the *Bos taurus* genome [32] was used as a reference. For each sample, at least 93% of reads were found to map to the reference genome and the alignments were evaluated for quality using TopHat [33], a splice-aware aligner, with default settings. Following alignment to the reference genome, the read count for each transcript was quantified using featureCounts [34].

Quality control on the raw reads and mapped sequence data was performed using FastQC [31], Multi QC [35] and RSeQC [36]. The quality of the data met accepted standards for downstream data analysis based on the mean quality scores and per sequence GC content.

## Identification of differentially expressed genes, functional enrichment and pathway analysis

Initially three RNA-Seq analysis algorithms, all within the Bioconductor R package, were used to identify differentially expressed genes (DEGs): CuffDiff [37], DESeq2 [38] and EdgeR [39]. Subsequent analysis of individual read counts across all samples indicated that results from the DESeq2 algorithm generated a dataset of intermediate size (n = 837 DEGs), relative to the other two algorithms (Edge R, n = 464; CuffDiff, n = 995) and this was used in subsequent downstream analysis. Pathway enrichment analysis was performed using the Kyoto Encyclopedia of Genes and Genomes (KEGG) database and Gene Ontology (GO) database DAVID version 6.7 (https://david.ncifcrf.gov). GO terms associated with adjusted P values below 0.05 were deemed statistically significant, after Benjamini-Hochberg adjustment. Similarly, for KEGG analysis, pathways with a P value below 0.05 were considered significant.

Additional pathway analysis of DEGs identified by DESeq2 was carried out using Ingenuity Pathway Analysis (IPA) software (QIAGEN; IPA® Systems_V2020, www.ingenuity.com). The Ensembl gene IDs and stable gene symbols for all genes in the dataset were uploaded to IPA along with DESeq2 expression values. This allowed IPA to automatically convert all *Bos taurus* IDs into a generic mammalian framework for analysis to identify 'top' ranking networks, Canonical Pathways, Upstream regulators and Diseases & Functions. In each case, Fisher's exact test was used to calculate a P value to check for enrichment of DEGs.

## Statistical significance of the overlap between TashAT2 and infection associated DEGs, and Sahiwal vs Holstein DEGs

To identify the relationship between previously identified DEGs associated with infection (IA) [20] and the current TashAT2 RNASeq2 dataset, we utilised the total number of genes that were commonly identifiable both in microarray (IA) and RNA-Seq (n = 17,555 genes). To test for statistical significance of the overlap, we used a Monte Carlo simulation to calculate the overlap expected to occur by chance [40, 41]. A chi-squared test was then used to statistically test whether the observed overlap was greater than that expected by chance. The results were also validated by calculating the representation factor, as described previously [28], where the P value was calculated as a normal approximation of the exact hypergeometric probability.

## Validation of differential expression of candidate genes by qRT-PCR

A subset of 12 putatively DEGs identified by RNA-Seq was selected for validation using qRT-PCR. Six genes with evidence of elevated expression in TashAT2 expressing BoMacs (pKP38) relative to non-expressing reverse orientation control lines (pKP37) were selected along with six which showed reduced relative expression. Primers were designed (see S1 File) and qRT-PCR performed, as described previously [17]. Briefly, 500 ng of total RNA for each sample was used for complementary DNA (cDNA) synthesis according to the manufacturer's instructions, using the Affinity Script cDNA Synthesis Kit and oligo (dT) primers provided with the kit (Stratagene, USA; Cat no. 600559). 1 μL cDNA for each sample was then used as template for qRT-PCR, using Brilliant III Ultra-fast SYBR®Green qPCR Master mix (Stratagene, 600548, Agilent Technologies) and the Stratagene Mx3005P system. Quantitative analysis across samples was performed using Stratagene MxPro Software. GAPDH was utilised as a housekeeping control gene based on previously published results of microarray experiments [17, 20]. The delta-delta CT method was then employed to calculate fold-change (vs the lowest expressed sample) and mean group differences (pKP38 vs pKP37) tested for significance (p < 0.05) by the Student's t-test.

## Generation of CRISPR-Cas9, TashAT2 knock-out lines

Oligo design, annealing and phosphorylation: F170_CR_G2_TashAT2_FW, F171_CR_G2_TashAT2_REV, F178_CR_G1_TashAT2_FW and F179_CR_G1_TashA-T2_REV guide sequences (see S1 File) were designed with the tool http://crispor.tefor.net [42], and then verified against the *Bos taurus* genome as non-targets for bovine genes. Primers were synthesised under commercial contract, https://www.microsynth.com. Guide sequences were annealed and phosphorylated in a 10 μL reaction: 1 μL forward primer (100 μM), 1 μL reverse primer (100 μM), 1 μL 10× T4 Ligation Buffer (New England Biolabs, USA), 6.5 μL ddH$_2$O, 0.5 μL T4 PNK (New England Biolabs, USA). Polymerase Chain Reaction was then performed: 30min at 37˚C, 5 min at 95˚C, ramped down to 25˚C for 5˚C/min (0.1˚C/sec).

Vector digestion and dephosphorylation: Plasmid pCRISPR_v3.1 was restriction digested with 1 μL *BbsI*, 5 μL 10X NEBuffer 2.1 (New England Biolabs, USA), 1 μg pCRISPR_v3.1 made up to a 50 μL reaction with ddH$_2$O and then incubated at 37˚C for 3 h, followed by a heat inactivation at 65˚C for 20 min. Digested plasmid DNA was then dephosphorylated using Calf Intestinal Phosphatase (CIP, New England Biolabs, USA): CIP 0.1 μL (1 unit), 1 pmol of DNA ends, 2 μL CutSmart Buffer 10× (New England Biolabs, USA), in a 20 μL reaction volume with ddH$_2$O and incubated at 37˚C for 30 min. The dephosphorylated plasmid DNA was cleaned using a PCR clean-up kit (Macherey-Nagel™ NucleoSpin™ Gel and PCR Clean-up Kit, Thermo-Fisher Scientific, UK).

Ligation of digested vector and annealed primers and transformation: 1 μL annealed oligo diluted 1:200, 1 μL T4 ligase 10× buffer, 0.5 μL T4 ligase (Promega, 1.5 units), 50 ng digested pCRISPR_v3.1 *BbsI* were made up to 10 uL with ddH$_2$O. Ligation was performed overnight at 16˚C. NEB C2987 bacteria (New England Biolabs, USA) were transformed with the pCRISPR_v3.1+guide plasmid (12 μL bacteria + 1 μL pCRISPR_v3.1+guide ligation) by incubating for 30 min on ice, 45 sec at 42˚C, 3 min on ice. 500 μL SOC media was then added, cells incubated for 1 h at 37˚C and plated out. Large-scale Endo-Free (Qiagen, UK) DNA preps were prepared and used for transfection.

The TashAT2 expressing BoMac cell line pKP38 was electroporated with plasmid 1193_TashAT2_G1_pCRISPR-CMV-Cas9-puroR_v3.1 or 1189_TashAT2_G2_p-CRISPR-CMV-Cas9-puroR_v3.1, as described above. Following 24 h recovery in standard medium, puromycin (InvivoGen, Europe) was added to 2.0 μg/mL and selection carried out for 3.5 d before continuing culture in fresh medium without puromycin. Transfected lines pKP38-CRISPR.g1 and pKP38-CRISPR.g2 were established by continued culture for 12 d. The two lines were then cloned by limiting dilution: cells were trypsinised, centrifuged (300 × g), resuspended in 2 mL medium and passed through a 21 g needle (×15) to generate a single cell suspension. Cells were counted and diluted down in culture medium to a concentration of 1 cell/100 μL and plated into 48 well plates. Plates were assessed for cell growth after 10 d and wells containing single colonies selected for expansion. Expanded clones were then screened by immunofluorescence using anti-TashAT2 antisera [22] relative to the pKP38 line. pKP38-CRISPR.g2 clones negative for TashAT2 expression were then validated by immunoblotting: no pKP38-CRISPR.g1 knock-out lines were obtained.

## Ethics statement

This study did not require ethical approval since it was conducted on *in vitro* cell lines that were established *ex vivo* from animals in a previous study where ethical approval was met [43] and no further human or animal participation was needed.

## Results

### Isolation of transfected cell lines and validation of TashAT2 expression

To generate a set of samples for robust analysis of bovine gene expression changes associated with the TashAT2 factor of *T. annulata*, six independent, stably transfected BoMac cell lines were generated. Three lines were transfected with the trans-gene in the correct orientation to allow expression of the TashAT2 protein fused to a V5 epitope tag (TashAT2 (pKP38) lines). Additionally, three negative control (pKP37) lines had the gene in the reverse orientation and could therefore not express TashAT2. To validate that expression of the transgene in these lines was as predicted, IFA was performed using an antibody against the V5 tag and a poly-specific antisera raised against a TashAT2 fusion protein [22]. Strong reactivity with either the anti-V5 antibody or the anti-TashAT2 serum was only detected in the lines transfected with the transgene in the correct orientation (Fig 1). This reactivity clearly co-located with the nucleus of the cell, as described previously for TashAT2 [22]. It was concluded that the six cell lines were suitable for analysis of TashAT2-modulated bovine gene expression by RNA-Seq.

### Identification of differentially expressed bovine genes in TashAT2 BoMac cells

To identify genes differentially expressed between the three TashAT2 BoMac cell lines and their corresponding negative controls, RNA-Seq was performed. Initial analysis showed that over 93% of sequence reads for all samples mapped to the bovine reference genome (UMD3.1). Normalised read counts were generated and, using the DESeq2 algorithm, differentially expressed genes (DE) were identified. Correlation across normalised counts among the replicates for negative control and TashAT2 samples was quantified by Pearson's correlation coefficient. All replicates within either group generated pair-wise coefficients of at least 0.95, indicating low variability among the replicates for both control and test datasets. To assess whether there was clear distinction between the TashAT2 and control bovine expression datasets, Principal Component Analysis (PCA) was carried out using the normalised read counts. The results (Fig 2) showed a clear separation between TashAT2 and control samples. The groups could be distinguished on the basis of the first component of the PCA, which explained 55% of the overall variance in the dataset. A greater level of variability was noted within the TashAT2 group compared to the control group on the basis of the second component. The analysis indicated that the three TashAT2 expressing lines compared to the three BoMac control lines had distinct DEG profiles.

The results obtained with DESeq2, using an adjusted P value with the cut-off set at 0.05, gave a total of 837 genes displaying differentially expressed transcript levels for the pKP38 vs pKP37 lines, thereafter denoted as the TashAT2/BoMac-DE. For the majority of genes in the dataset, differential read count values were consistent (i.e. higher or lower in all TashAT2 samples relative to control). Of the 837 DEGs, 319 were higher (with a maximum fold change of 2.4 $\log_2$), while 518 genes displayed lower levels of transcripts in TashAT2 expressing cell lines (maximum fold change, 2.58 $\log_2$). These results are summarised by volcano plot (S1 Fig), with the full list of TashAT2/BoMac-DE genes given in S2 File.

Tables 1 and 2 show the top 20 DEGs in TashAT2 BoMac lines relative to control lines. Even within this limited list of genes, a number possess gene ontology (GO) annotation indicating they could play a role in modulating the phenotype of BoMac cells and, potentially, perform a similar role in *T. annulata* infected leukocytes. For genes displaying lower expression level in TashAT2 BoMac, a number have been associated with tumour suppression activity. These include the *CYGB* gene encoding cytoglobulin, which is a known inhibitor of mitosis

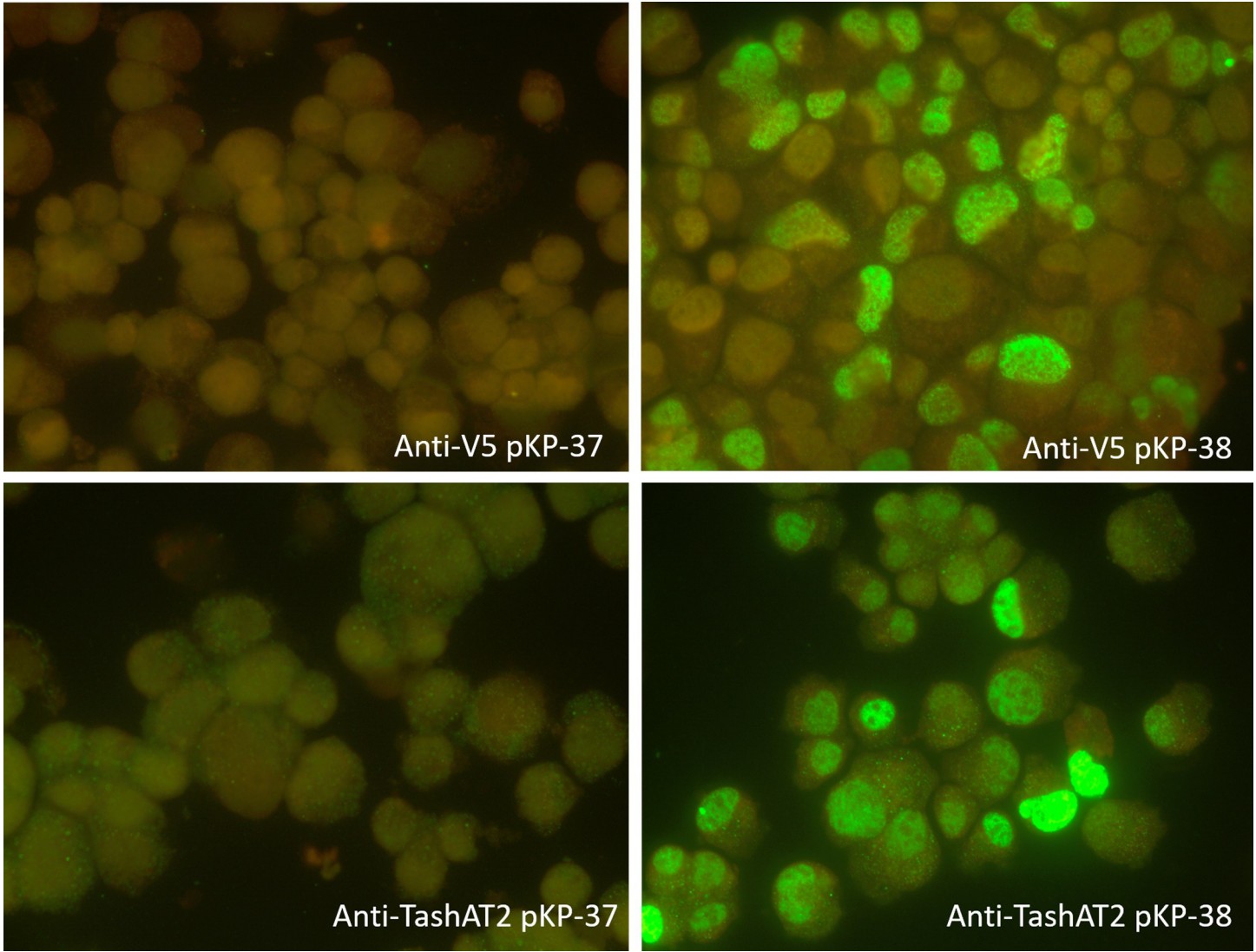

**Fig 1. Validation of TashAT2 expression in stably transfected (pKP38) BoMac cells.** Reactivity against TashAT2 was detected by immunofluorescence using an anti-V5 monoclonal antibody (top row) and an anti-TashAT2 polyspecific Ab (bottom-row) in BoMac pKP-38 (TashAT2 expressing) cells. Strong reactivity was limited to the nuclear region of positive cells. There was no reactivity in the control BoMac pKP-37 cells.

[44] and the calcium/calmodulin dependent protein kinase II inhibitor 1 gene (*CAMK2N1*) that can act as a tumour suppressor and block metastasis [45]. Likewise, the genes encoding the retinoic acid receptor responder 2 (RARRES2) and G protein subunit gamma 4 (GNG4) are thought to function as tumour suppressors [46, 47]. A number of genes encoding nuclear factors that modulate gene expression were also among the top 20 down-regulated genes. These included, interferon regulatory factor 6 (IRF6) and HOXA10 reported to function in alternative macrophage activation [48] and haematopoietic cell lineage commitment [49], respectively. In contrast, genes showing higher expression were more associated with products that have a surface/membrane location or are predicted to function in signal transduction. The gene encoding GULP1, for example, has been found to be involved in phagocytosis, integrin signaling [50] and TGFβ signaling [51]. In addition, Adhesion G Protein-Coupled Receptor G1 (ADGRD1) functions in cell matrix adhesion and cancer progression [reviewed in 52], and

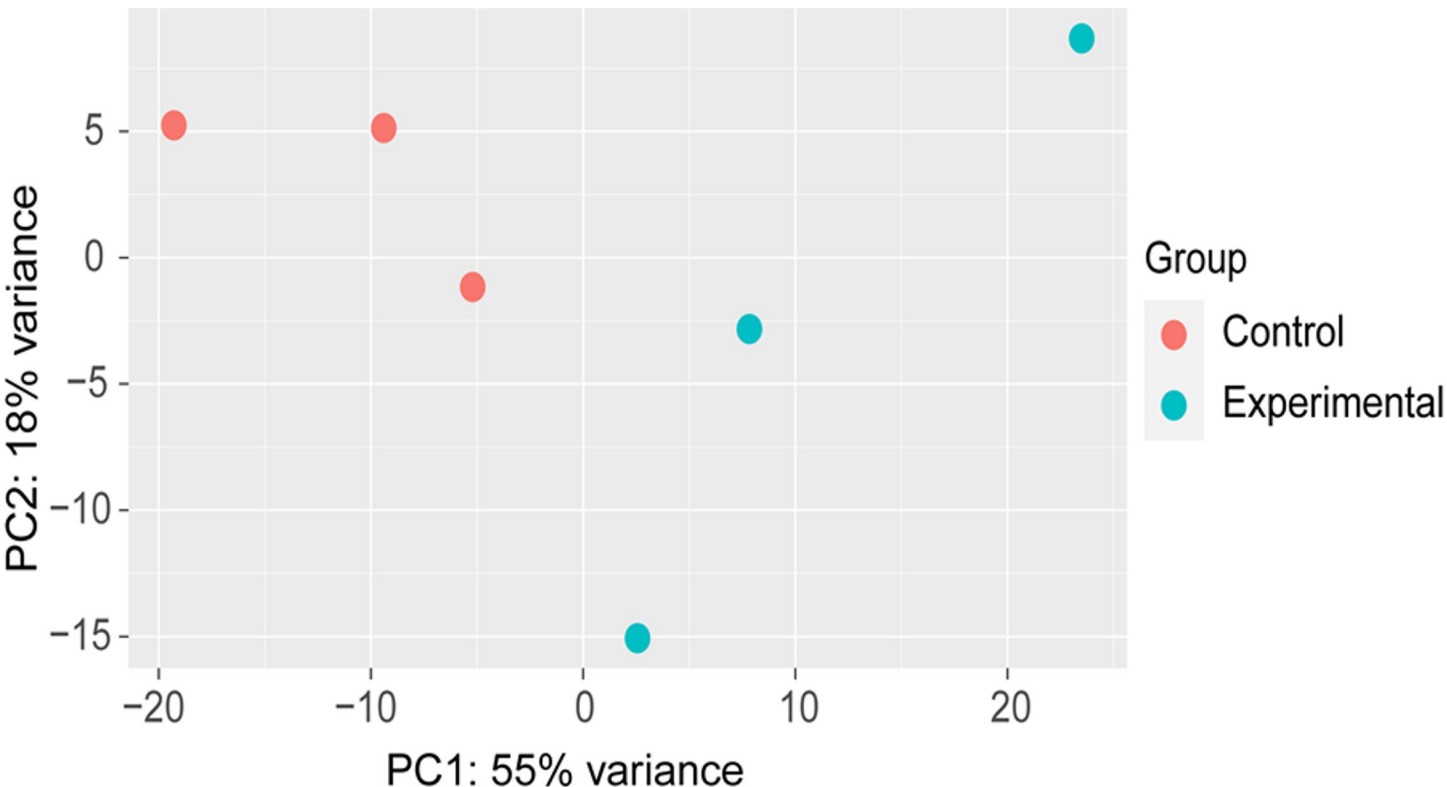

**Fig 2. Clustering of samples by TashAT2-expression.** A PCA plot, showing clustering based on the TashAT2/BoMac-DE dataset derived from three Control cell lines (pKP37, red) and three Experimental TashAT2 expressing cell lines (pKP38, blue). The plot was generated from the variation among normalised read counts.

*KITLG* (SCF/c-Kit) encodes the cytokine that acts as the ligand for the tyrosine-kinase KIT receptor. KITLG has an essential role in the regulation of cell survival, cell differentiation, proliferation, haematopoiesis and mediation of cell migration, operating via activation of PI3K and Akt [53, 54]. Genes encoding ISG15 and secreted phosphoprotein 1/Osteopontin (SPP1), previously identified as expressed at a lower level in TashAT2/BoMac lines [19, 55], showed lower mean RNA-Seq expression values in the current study. However lower expression was only indicated as statistically significant for *SSP1*, due to variance in expression values across individual samples for *ISG15*.

## Pathway analysis indicates modulation of genes involved in adhesion and oncogenesis

Pathway analysis offers a powerful approach for identification, from transcriptomic data, of gene networks and regulatory interactions. As the number of genes in the TashAT2/BoMac-DE dataset was moderately large, we tested for enrichment of pathways using the Kyoto Encyclopedia of Genes and Genomes database (KEGG) and Ingenuity Pathway Analysis (IPA). Thirteen KEGG pathways were found to display significant (P < 0.05) enrichment for genes within the TashAT2/BoMac-DE dataset (S3 File). The top ten pathways are illustrated in Fig 3A. The top two pathways implicate involvement of TashAT2/BoMac-DE genes in the related functions of "Focal adhesion" and "ECM (Extracellular Matrix) interaction" via modulated expression of collagens, integrins, thrombospondins and fibronectin (S3 File). These results were supported by IPA canonical pathways (Fig 3B) where the most significant enrichment was for "ILK (Integrin-linked kinase) signaling" for which a negative Z-score (-1.8) was

**Table 1. Top 20 genes (by fold change) displaying reduced expression in TashAT2 BoMac lines relative to control lines.**

| Ensembl Gene ID | Entrez Gene Name | Gene symbol | Location | Type(s) | DESeq2 | | | | | |
|---|---|---|---|---|---|---|---|---|---|---|
| | | | | | baseMean | log₂ FC | lfcSE | stat | pvalue | padj |
| ENSBTAG00000038662 | Gap junction protein beta 6 | GJB6 | Plasma Membrane | transporter | 121.44 | -2.59 | 0.25 | -10.16 | 2.93E-24 | 1.04E-20 |
| ENSBTAG00000005556 | Cytoglobin | CYGB | Cytoplasm | transporter | 173.73 | -2.15 | 0.28 | -7.71 | 1.31E-14 | 1.39E-11 |
| ENSBTAG00000021928 | Divergent protein kinase domain 1B | DIPK1B | Other | other | 78.57 | -2.04 | 0.28 | -7.32 | 2.53E-13 | 1.74E-10 |
| ENSBTAG00000025622 | DEF6 guanine nucleotide exchange factor | DEF6 | Extracellular Space | other | 547.34 | -2.02 | 0.27 | -7.52 | 5.49E-14 | 5.42E-11 |
| ENSBTAG00000040082 | Homeobox A10 | HOXA10 | Nucleus | transcription regulator | 45.61 | -2.00 | 0.28 | -7.16 | 8.19E-13 | 4.92E-10 |
| ENSBTAG00000004792 | Charged multivesicular body protein 4C | CHMP4C | Cytoplasm | other | 97.72 | -1.96 | 0.28 | -7.04 | 1.87E-12 | 9.22E-10 |
| ENSBTAG00000026896 | Forkhead box F2 | FOXF2 | Nucleus | transcription regulator | 56.23 | -1.92 | 0.28 | -6.89 | 5.48E-12 | 2.37E-09 |
| ENSBTAG00000007665 | Natriuretic peptide receptor 3 | NPR3 | Plasma Membrane | G-protein coupled receptor | 2620.51 | -1.88 | 0.19 | -10.16 | 3.00E-24 | 1.04E-20 |
| ENSBTAG00000018123 | Fibulin 5 | FBLN5 | Extracellular Space | other | 1013.97 | -1.87 | 0.25 | -7.40 | 1.41E-13 | 1.14E-10 |
| ENSBTAG00000009156 | Calcium/calmodulin dependent protein kinase II inhibitor 1 | CAMK2N1 | Plasma Membrane | kinase | 219.06 | -1.83 | 0.28 | -6.57 | 5.03E-11 | 1.69E-08 |
| ENSBTAG00000021978 | Parvin beta | PARVB | Cytoplasm | other | 32.41 | -1.73 | 0.28 | -6.23 | 4.75E-10 | 1.23E-07 |
| ENSBTAG00000023472 | Protein phosphatase 1 regulatory inhibitor subunit 14A | PPP1R14A | Cytoplasm | phosphatase | 329.84 | -1.72 | 0.24 | -7.05 | 1.75E-12 | 8.95E-10 |
| ENSBTAG00000007678 | Mohawk homeobox | MKX | Nucleus | transcription regulator | 215.66 | -1.71 | 0.28 | -6.11 | 9.90E-10 | 2.24E-07 |
| ENSBTAG00000002849 | Interferon regulatory factor 6 | IRF6 | Nucleus | transcription regulator | 128.22 | -1.69 | 0.28 | -6.05 | 1.45E-09 | 3.14E-07 |
| ENSBTAG00000020568 | NK2 homeobox 5 | NKX2-5 | Nucleus | transcription regulator | 77.62 | -1.68 | 0.28 | -6.05 | 1.44E-09 | 3.14E-07 |
| ENSBTAG00000004215 | Retinoic acid receptor responder 2 | RARRES2 | Plasma Membrane | transmembrane receptor | 146.25 | -1.64 | 0.24 | -6.77 | 1.31E-11 | 5.02E-09 |
| ENSBTAG00000017599 | Nuclear receptor subfamily 2 group F member 1 | NR2F1 | Nucleus | ligand-dependent nuclear receptor | 2733.41 | -1.61 | 0.19 | -8.62 | 6.66E-18 | 1.31E-14 |
| ENSBTAG00000002908 | G protein subunit gamma 4 | GNG4 | Plasma Membrane | enzyme | 622.89 | -1.60 | 0.27 | -5.94 | 2.83E-09 | 5.65E-07 |
| ENSBTAG00000011207 | Calponin 1 | CNN1 | Cytoplasm | other | 7992.64 | -1.59 | 0.21 | -7.74 | 9.99E-15 | 1.15E-11 |
| ENSBTAG00000013227 | Snail family transcriptional repressor 2 | SNAI2 | Nucleus | transcription regulator | 308.84 | -1.59 | 0.25 | -6.34 | 2.34E-10 | 6.67E-08 |

computed, predicting a decreased activation state for this pathway, although this was not denoted as significant (i.e. was not < 2). The prediction of reduced activity for this pathway is illustrated (S2 Fig) with the majority of genes displaying reduced expression values (green) thus influencing cell motility, cell adhesion and cytoskeletal reorganisation.

The third most significant KEGG pathway was the PI3K/Akt pathway, which was notable as it is known to be activated in *T. annulata* infected leukocytes [56]. Genes in this pathway included the integrins and collagen genes displaying lower relative levels of expression in

**Table 2. Top 20 genes (by fold change) displaying elevated expression in TashAT2 BoMac lines relative to control lines.**

| Ensembl Gene ID | Entrez Gene Name | Gene symbol | Location | Type(s) | DESeq2 | | | | | |
|---|---|---|---|---|---|---|---|---|---|---|
| | | | | | baseMean | log₂ FC | lfcSE | stat | pvalue | padj |
| ENSBTAG00000003088 | LanC like 3 | LANCL3 | Other | other | 122.22 | 2.45 | 0.24 | 10.18 | 2.47E-24 | 1.04E-20 |
| ENSBTAG00000007141 | GULP PTB domain containing engulfment adaptor 1 | GULP1 | Cytoplasm | other | 238.12 | 2.44 | 0.24 | 10.33 | 5.16E-25 | 7.12E-21 |
| ENSBTAG00000048246 | Dipeptidyl peptidase 4 | DPP4 | Plasma Membrane | enzyme | 190.89 | 2.08 | 0.28 | 7.46 | 8.82E-14 | 8.12E-11 |
| ENSBTAG00000015330 | Uncharacterised protein (UP) | - | - | - | 221.87 | 1.95 | 0.28 | 6.99 | 2.78E-12 | 1.28E-09 |
| ENSBTAG00000006546 | Glutathione s-transferase alpha 1 | GSTA2 | Cytoplasm | enzyme | 231.87 | 1.92 | 0.26 | 7.42 | 1.16E-13 | 9.98E-11 |
| ENSBTAG00000045757 | Troponin c1, slow skeletal and cardiac type | TNNC1 | Cytoplasm | other | 2090.73 | 1.89 | 0.21 | 8.98 | 2.81E-19 | 6.46E-16 |
| ENSBTAG00000018497 | Caveolae associated protein 2 | CAVIN2 | Plasma Membrane | other | 195.79 | 1.83 | 0.26 | 7.01 | 2.34E-12 | 1.11E-09 |
| ENSBTAG00000015317 | Solute carrier family 4 member 10 | SLC4A10 | Plasma Membrane | transporter | 33.65 | 1.76 | 0.28 | 6.34 | 2.37E-10 | 6.67E-08 |
| ENSBTAG00000004720 | Sulfatase 1 | SULF1 | Cytoplasm | enzyme | 3548.01 | 1.74 | 0.26 | 6.68 | 2.39E-11 | 8.46E-09 |
| ENSBTAG00000033726 | Glutamate receptor interacting protein 1 | GRIP1 | Plasma Membrane | transcription regulator | 62.24 | 1.73 | 0.28 | 6.20 | 5.47E-10 | 1.37E-07 |
| ENSBTAG00000017549 | Kit ligand | KITLG | Extracellular Space | growth factor | 2664.25 | 1.69 | 0.23 | 7.25 | 4.17E-13 | 2.62E-10 |
| ENSBTAG00000016137 | Zinc finger protein 608 | ZNF608 | Other | other | 261.20 | 1.65 | 0.24 | 6.95 | 3.56E-12 | 1.59E-09 |
| ENSBTAG00000015885 | Protein tyrosine phosphatase, receptor type, f polypeptide (ptprf), interacting protein (liprin), alpha 4 | PPFIA4 | Plasma Membrane | other | 2054.57 | 1.64 | 0.25 | 6.53 | 6.55E-11 | 2.15E-08 |
| ENSBTAG00000013848 | Adhesion g protein-coupled receptor d1 | ADGRD1 | Plasma Membrane | G-protein coupled receptor | 1073.87 | 1.61 | 0.27 | 5.97 | 2.44E-09 | 4.95E-07 |
| ENSBTAG00000014069 | Pyruvate dehydrogenase kinase 4 | PDK4 | Cytoplasm | kinase | 76.80 | 1.61 | 0.27 | 5.86 | 4.71E-09 | 8.66E-07 |
| ENSBTAG00000011500 | Calsequestrin 2 | CASQ2 | Cytoplasm | other | 273.25 | 1.58 | 0.27 | 5.82 | 6.03E-09 | 1.06E-06 |
| ENSBTAG00000016140 | PPFIA binding protein 2 | PPFIBP2 | Nucleus | phosphatase | 546.67 | 1.57 | 0.26 | 6.11 | 9.69E-10 | 2.24E-07 |
| ENSBTAG00000010954 | ADP-ribosyltransferase 3 | ART3 | Plasma Membrane | enzyme | 163.61 | 1.56 | 0.28 | 5.59 | 2.25E-08 | 3.37E-06 |
| ENSBTAG00000037399 | Uncharacterised protein (UP) | - | - | - | 1116.99 | 1.49 | 0.27 | 5.53 | 3.23E-08 | 4.51E-06 |
| ENSBTAG00000011634 | Acid phosphatase 3 | ACP3 | Extracellular Space | phosphatase | 118.21 | 1.48 | 0.28 | 5.36 | 8.12E-08 | 1.01E-05 |

PKP38 lines, while genes encoding (macrophage) colony stimulating factor (*CSF1*), *KITLH*, forkhead box 3 (*FOXO3*), nerve growth factor response (*NFGR*), *PIK3R3* and *PRKAA2* all showed higher expression in pK38 lines compared to the control pKP37 lines (S3 File). While the PI3K/Akt pathway in IPA was below the significance threshold (-log(p-value) 1.32; Z-score 0), a number of its genes also form part of the ILK signaling pathway (*PIK3R3*, *PPP2CB*, *PPP2R3A*, *PPP2R5A*), and within the ILK pathway, *PI3K* was predicted to be up-regulated (see S2 Fig).

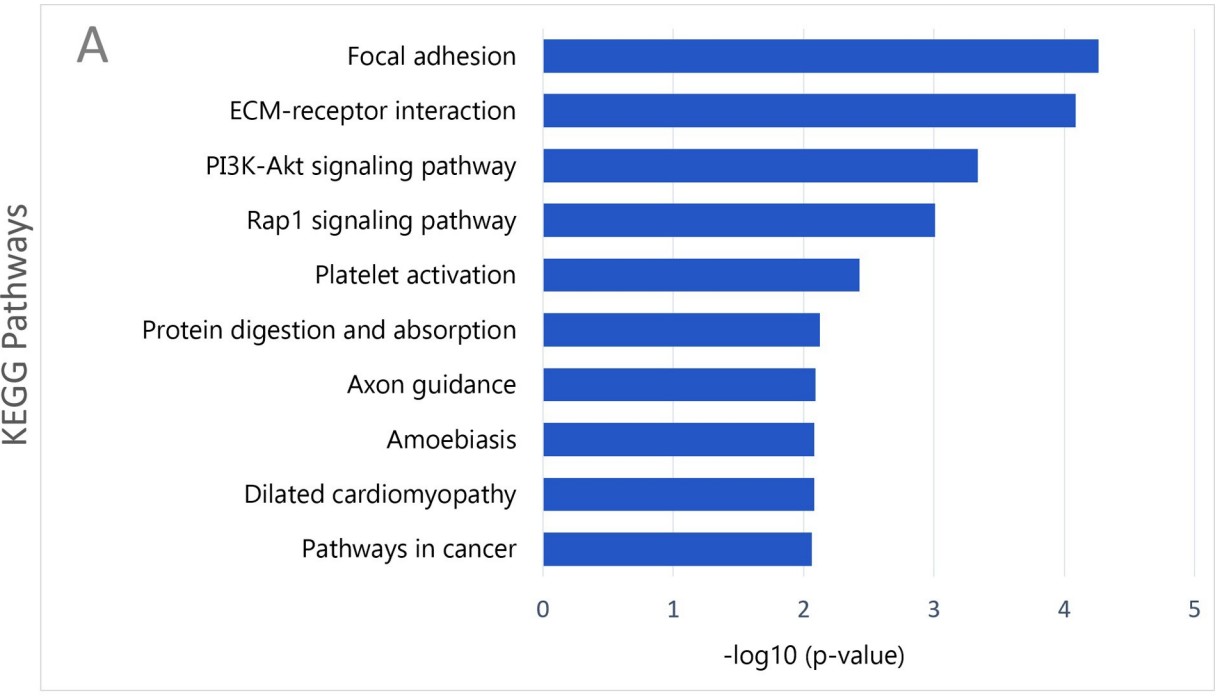

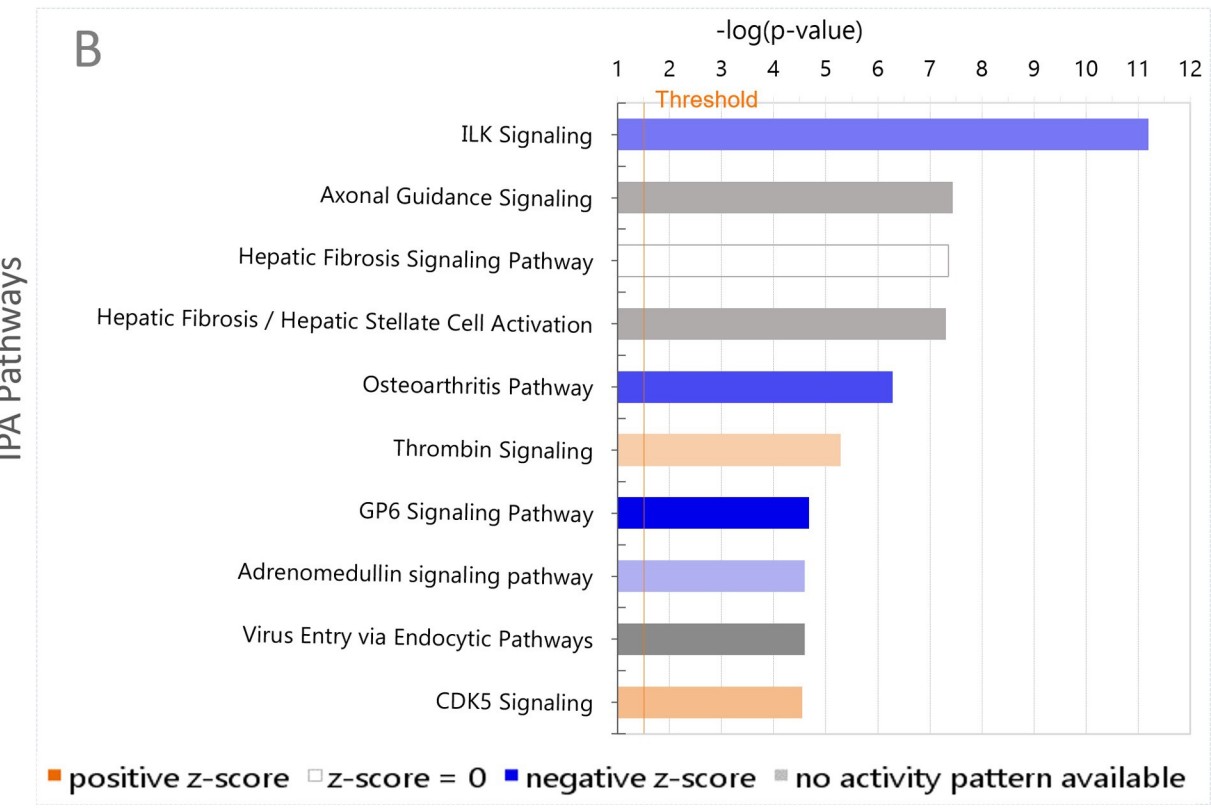

**Fig 3. KEGG and IPA pathways enriched in the TashAT2/BoMac-DE genes set (n = 837).** (A) Top 10 significant KEGG pathways as determined by hypergeometric test. P-values were adjusted by the Benjamini-Hochberg method (P < 0.05) and for FDR (P < 0.05). DEGs were mapped to KEGG pathways using the STRING Database and DAVID Bioinformatics Suite. For each KEGG pathway, the x-axis bar shows the– $\log_{10}$ (p-value) after Benjamini-Hochberg correction. (B) Top 10 Canonical pathways identified by IPA are indicated: each bar represents–log (p-value) for over-representation of the modulated genes in the selected pathway. Threshold (orange line) denotes p = 0.05.

The pathway "Axon guidance" displayed significant enrichment in both KEGG and IPA. Genes displaying higher expression in TashAT2 transfected BoMacs (pKP38) included Slit guidance ligand 2 (*SLIT2*) and *SLIT3*, while genes for netrins (*NTN1*, *NTN4*) and an integrin subunit (*ITGB1*) of a receptor for netrin [57] displayed lower relative expression. In the IPA "Axonal Guidance Signaling" pathway, a general activation status could not be predicted (S4 File). Two additional top ten pathways identified by IPA displayed significant negative Z-scores (<-2). These were "Osteoarthritis Pathway" (Z-score -2.4) and "GP6 Signaling Pathway" (Z-score -3.36). A previous study on osteoarthritis that highlighted this pathway also identified pathways for axon guidance signaling and GP6, as well as modulation of genes linked to cellular adhesion [58].

An association with oncogenesis was indicated using IPA Disease and Function analysis (Fig 4A and 4B). Thus, the most significant "Disease and disorders" was cancer (Z-score 0.379 with 782 molecules). Likewise, the majority of the most significant IPA "Molecular and Cellular Functions" could be linked to oncogenesis and cancer and included "Cellular Movement", "Cell Death and Survival", "Cellular Growth and proliferation", all of which had significant Z scores (Fig 4B and S5 File). "Cellular Assembly and Organization" and Cell Morphology" were also within the top 10 "Molecular and Cellular Functions" categories.

To determine if genes in the TashAT2/BoMac-DE dataset were associated with modulation by particular regulators, the upstream regulator analysis function in IPA was utilised. Of the ten most significant activated regulators, five were predicted to be miRNAs with let-7a-5p displaying the highest positive Z score, next to that of alpha catenin (Fig 5A). Among the most significant inhibited regulators, growth factors (e.g. TGFβ1, TGFβ3 and EGF) and transcriptional regulators (e.g. SMAD3, ATF4 and MTPN) were prominent with interferon alpha group cytokines, the SMAD3 transcription factor and the TGFβ1 growth factor showing the greatest negative Z-scores. Illustration of a predicted inhibition of SMAD3 activity, based on detected modulation of its target genes in the regulatory network, is shown in Fig 5B. Here the majority of target genes display reduced expression in the pKP38 lines, including integrins (*ITG1* and *ITG5*), collagens (*COL1A1*, *COL1A2*, *COL2A1* and *COL6A1*), vimentin and fibronectin 1 (*FN1*).

Other regulators predicted by the analysis (S6 File) were of interest due to existing knowledge of association with either *Theileria* infection or possession of an AT-hook motif. These included NF-κB complex (Z-score -0.8), IL-6 (-0.97), TP53 (-1.0), JunB (-1.69), Fos (-1.53), Akt (-1.3) and the mammalian AT-hook protein HMGA1 (0.07).

## A subset of TashAT2/BoMac-DE genes are modulated in infected host leukocytes

To test whether any of the genes modulated by *TashAT2* expression were also associated with changes that occur upon infection of bovine leukocytes with *T. annulata*, the TashAT2/BoMac-DE dataset was compared with a set of bovine genes demonstrated to display altered expression in infected TBL20 cells relative to the uninfected BL20 lymphosarcoma cell line [20]. Of the 837 TashAT2/BoMac-DEGs, 209 overlapped with this dataset (S7 File) and were, therefore, indicated as TashAT2-modulated and infection-associated (TashAT2/IA). The overlap was found to be significant (p < 0.001) by Monte Carlo simulation (Fig 6A) with a significant representation factor of 1.5 (P < $1.38 \times 10^{-10}$) calculated as described [59]. A number of the TashAT2/IA genes could influence the phenotype of the infected cell. For example, the histone deacetylase 9 (*HDAC9*) gene that displays lower relative expression in infected cells [20] was also expressed at a reduced level in TashAT2 transfected BoMac lines, as was the gene encoding synaptopodin 2 (*SYNPO2*), which induces focal adhesion [60] and can act as a

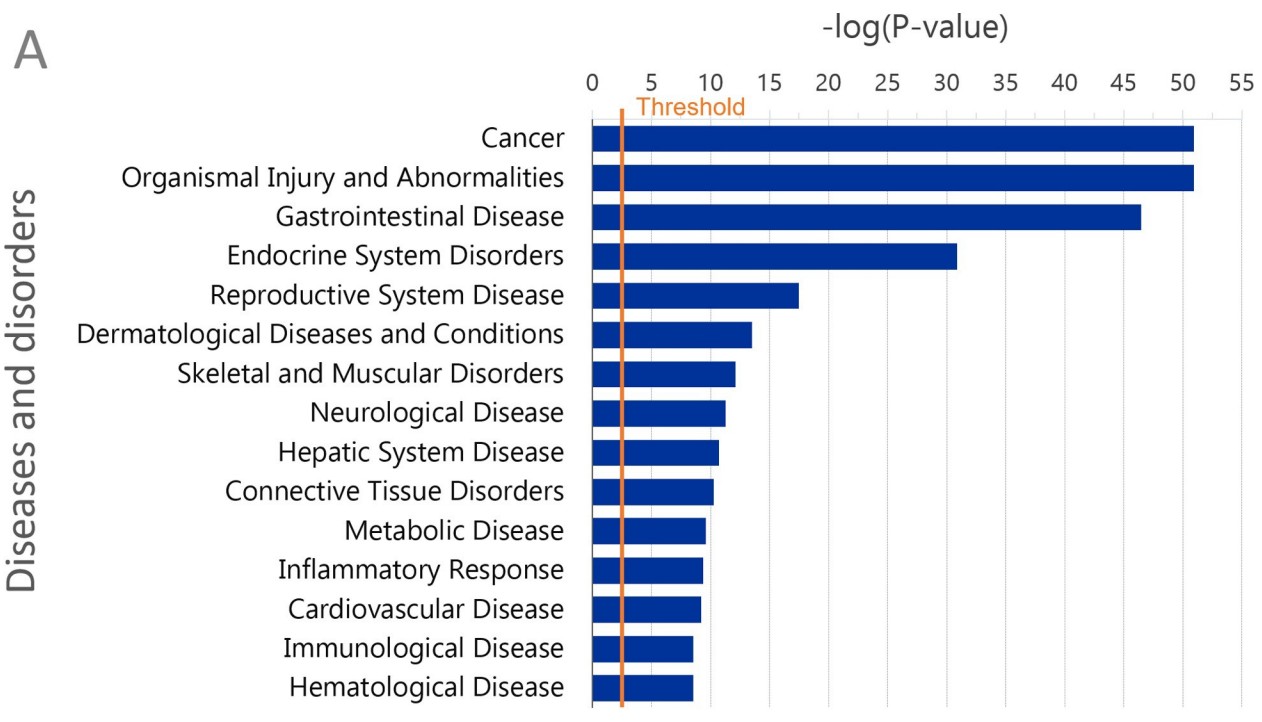

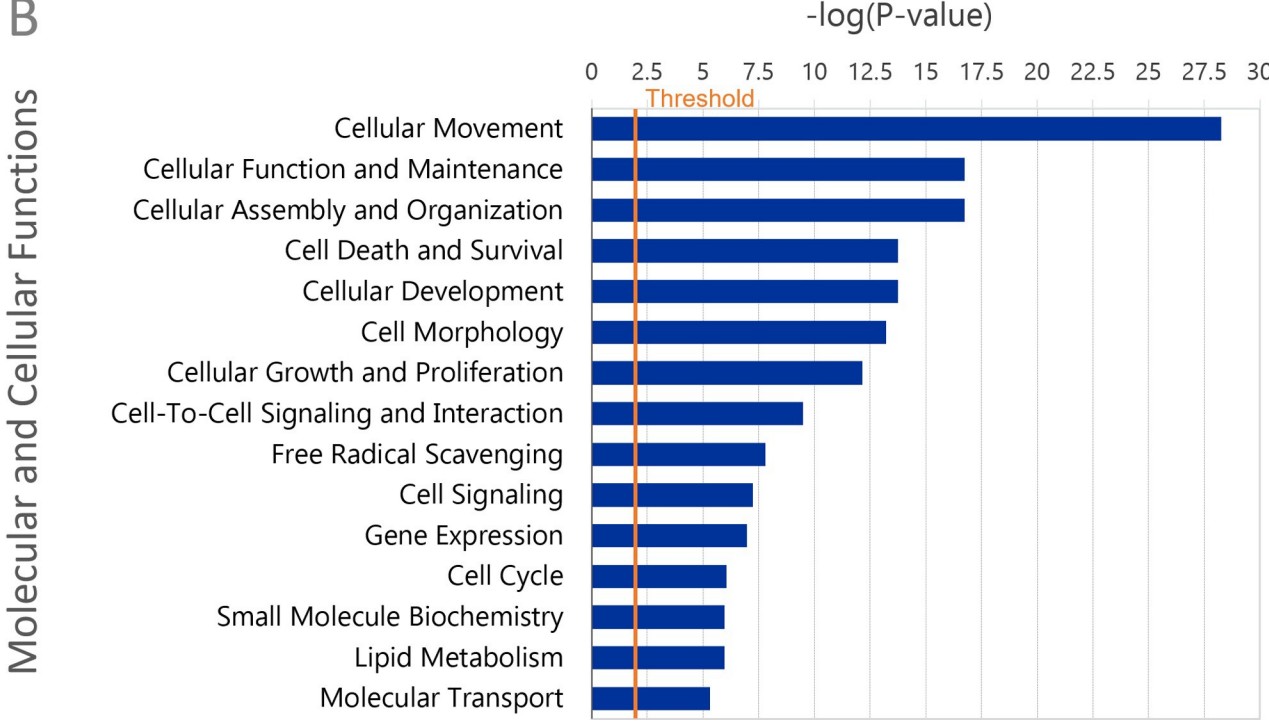

**Fig 4. Functional enrichment analysis of TashAT2/BoMac-DE dataset.** Top 15 categories in Diseases and disorders (A) and Molecular and Biological Functions (B) identified as significantly enriched in the dataset by IPA. Categories are shown on the y-axis with the significance of the enrichment as–log (p-value) denoted along the x-axis; the orange vertical line represents the threshold for significance (p = 0.05).

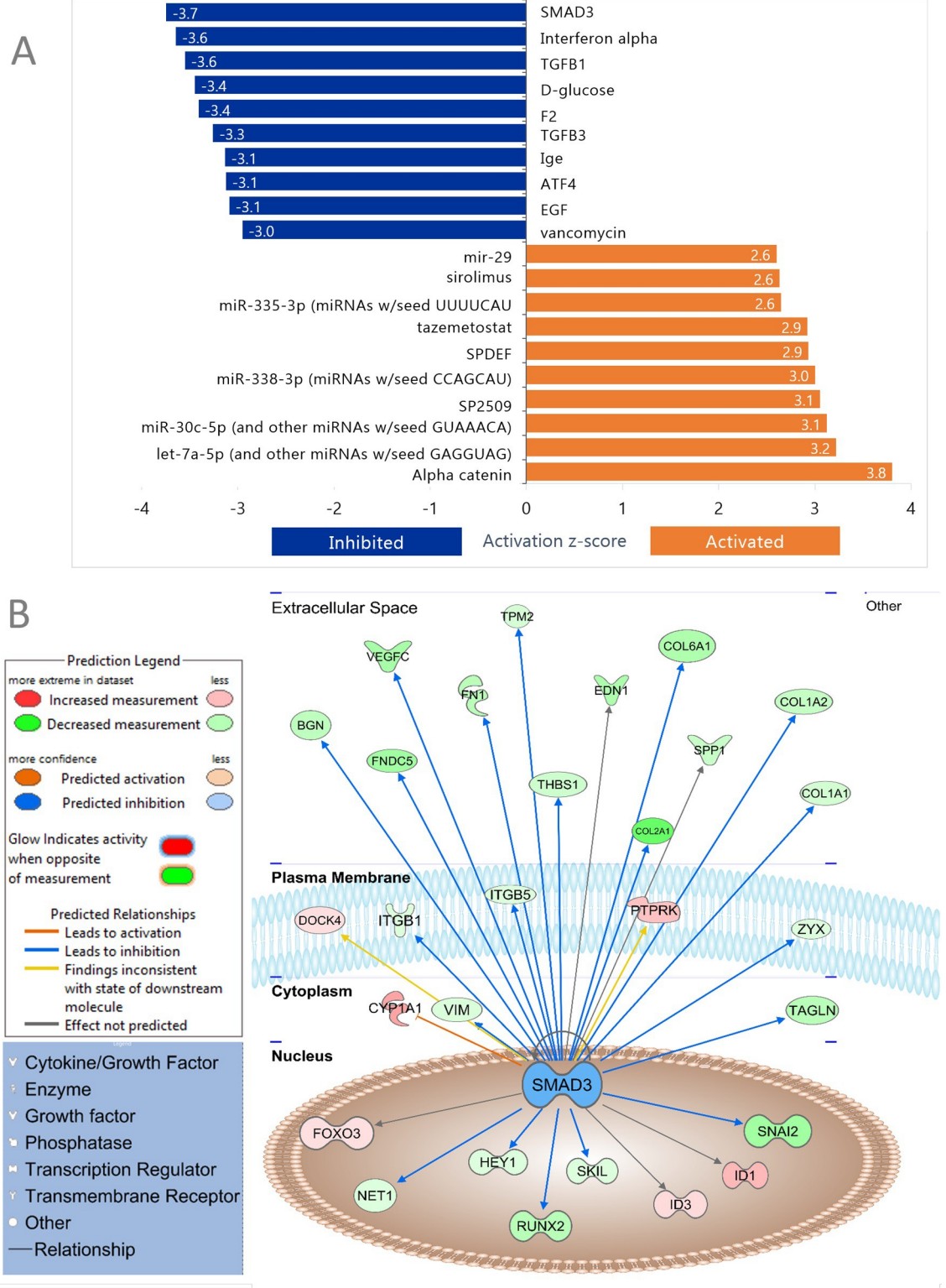

**Fig 5. IPA upstream regulatory network analysis of TashAT2/BoMac-DE dataset.** (A) Top 10 significantly (Z-score) altered upstream regulators are shown by horizontal bars based on Z-scores. Blue colour indicates inhibition while orange colour indicates activation. (B) SMAD3 regulatory network shown in subcellular mode. SMAD3 transcription factor is predicted to be inhibited in the TashAT2 modulated dataset (p = 9.96 × 10$^{-7}$, Z-score = -3.74). Blue arrows connecting the nodes show upstream regulators with a predicted inhibitory role on their downstream genes, an orange line indicates when upstream regulators have a predicted activation effect on their target gene, a yellow arrow shows that the findings of downstream genes are inconsistent with the prediction based on previous findings. The colours within individual nodes indicate the status and intensity of down (green) or up (red) modulated expression.

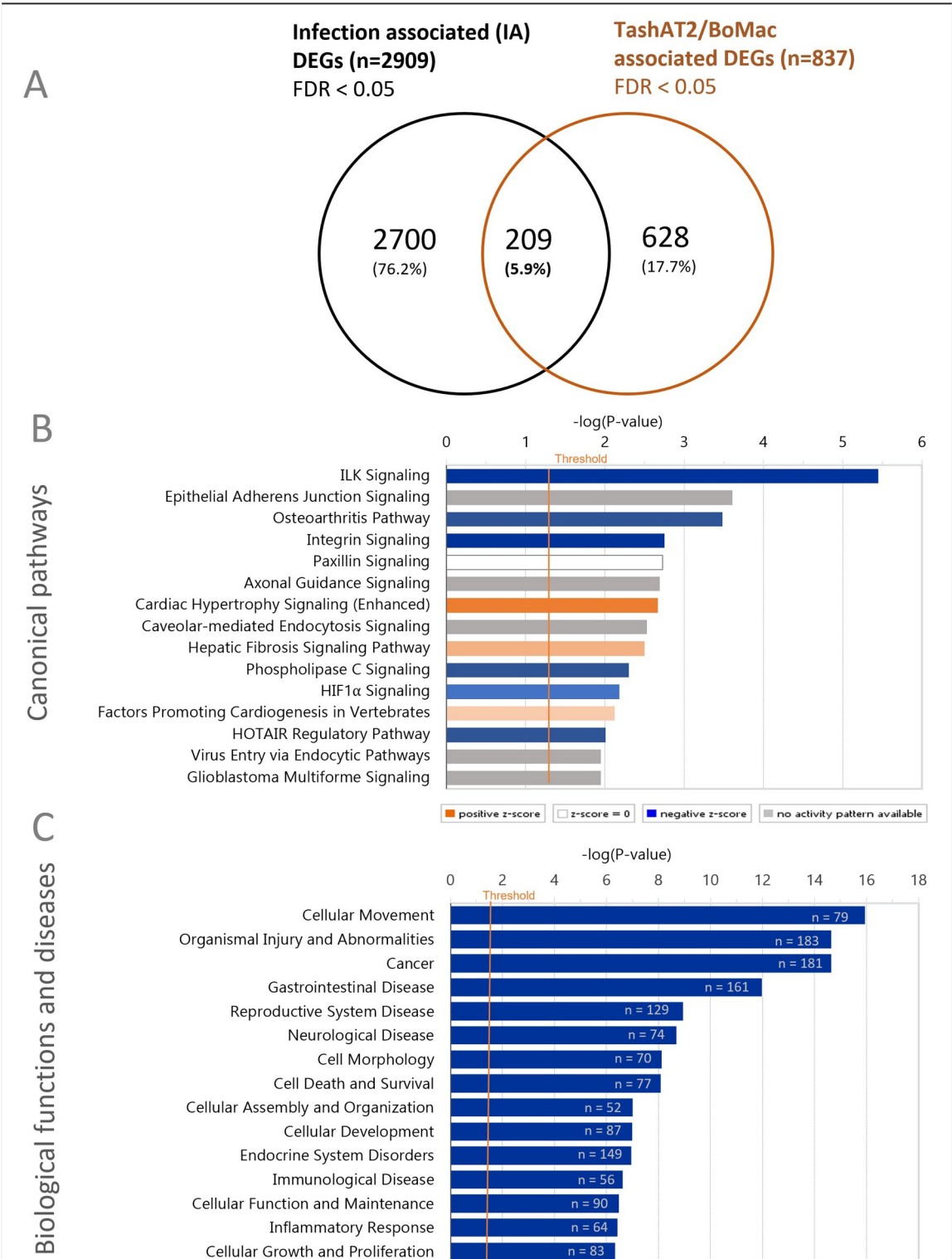

**Fig 6. IPA analysis of TashAT2/IA gene set.** (A) Genes overlapping across experimental conditions (TashAT2 expression or *Theileria* infection) are shown by Venn diagram. Monte Carlo simulation and Chi square analysis identified this overlap to be significant (p < 0.001). (B) Top 15 canonical pathways and (C) Top 15 Biological functions and diseases categories identified as significantly enriched by IPA in the TashAT2/IA dataset (n = 209). Individual categories are shown along the y-axis with the significance of the enrichment (–log transformed) denoted along the x-axis. Each bar represents:–log (p-value) for enrichment and number of genes in a category. The orange vertical line represents the threshold for significance (p = 0.05).

tumour suppressor [61]. Genes whose expression was elevated by infection and TashAT2 include *PPFIA4*, encoding PTPRF interacting protein alpha 4 (Liprin-alpha-4), whose expression is induced by hypoxia-inducible factor 1α (HIF-1α) and is thought to operate in cell to cell adhesion [62], and Rho GTPase activating protein 18 (*ARHGAP18)* that functions to suppress polymerisation of actin and regulate cell shape, spreading and cellular migration [63].

IPA analysis of the TashAT2/IA dataset generated similar results to those obtained with the full TashAT2/BoMac-DE dataset (compare Figs 3 and 4 with 6B and 6C) although, as might be expected, statistical significance was poorer and Z scores closer to zero. Thus, the pathways "ILK Signaling", "Osteoarthritis Pathway", "Paxillin Signaling", "Axonal Guidance" and "Cardiac Hypertrophy Signaling" were in the top 12 canonical pathways for both datasets, with a number of genes displaying the same directional change for TashAT2 and infection, such as the ILK pathway in which 6 of 10 genes (*ITGB5*, *ITGB7*, *LEF1*, *PARVB*, *VCL* and *VIM*) behave in the same manner (S7 and S8 Files). Notable differences in IPA of TashAT2/IA, relative to that of the TashAT2/BoMac-DE dataset, were the absence of the GP6 pathway, due to the lack of collagen (*COL*) genes, and the presence of the HIF1α pathway.

IPA for "Biological function and diseases" identified categories that could be associated with the oncogenic phenotype of the infected leukocyte including "Cellular movement", "Cancer" and "Cell death and survival" (Fig 6C). Within "Cellular Movement" 45 out of 79 genes (55.9%) showed the same direction of modulation in both the TashAT2-modulated and TBL20-modulated datasets, including *HDAC9*, *ITGB7*, *LEF1*, *PARVB*, *SLIT2*, *SYNPO2*, *THBS1* and *WNT10B*; while others *FBLIM1*, *ITGB1* and *LITAF* (lipopolysaccharide-induced TNF factor), displayed opposite directional change across these two datasets (S7 and S8 Files). Enrichment of genes within "Cell Morphology" and "Cellular Assembly and Organisation" was also of interest with subcategories "reorganisation of cytoskeleton", "morphology of cells", "shape changes" and "ruffling" highlighted (Fig 6C and S8 File).

Upstream regulator analysis performed for the TashAT2/IA dataset showed similarity to results obtained for the full TashAT2/BoMac-DE dataset, although with reduced significance and lower Z scores. Thus, FOS, IFNG, SMAD3, SNAI1 (also known as SLUGH2 or SNAIL) and TGFβ1 were identified for both datasets. Some predicted regulators gained significance with growth factors IL2, KITLG and TNF, and the transcription factor LMO2 showing higher ranked positions (Fig 7A). Modulated expression of *IL2* and *TNF* in *Theileria* infected cells has been reported previously [64, 65]. *LMO2* was not indicated as modulated by TashAT2 in our RNA-Seq analysis due to variance in one control line and *KITLG* was not identified as modulated by infection in the TBL20/BL20 dataset. However, our previous microarray study, using an independently generated TashAT2 transfected BoMac and control line, displayed reduced *LMO2* expression associated with TashAT2 expression. This result was validated by semi-quantitative PCR and lower *LMO2* expression demonstrated for infected TBL20 and the cloned D7 (*ex vivo*) cell line relative to uninfected BL20 [55]. Inhibited LMO2 activity is predicted to modulate expression of a number of important target genes in both TashAT2 transfected BoMacs and *Theileria* infected BL20 (12/13 predicted), the majority (8/12) of which show the same direction of change (see Fig 7B).

## qRT-PCR validation of RNA-Seq candidate genes modulated by TashAT2

To validate the RNA-Seq data, qRT-PCR analysis was performed with RNA isolated from the six TashAT2 transfected and control cell lines. Twelve genes were selected based on representation of different RNA-Seq profiles across the TashAT2/BoMac-DE dataset. The results are summarised in Fig 8. Genes encoding prostatic acid phosphatase (*ACPP*), MUM1 like 1 (*MUM1L1*, also known as *PWWP3B*) and von Willebrand factor A domain containing 5A

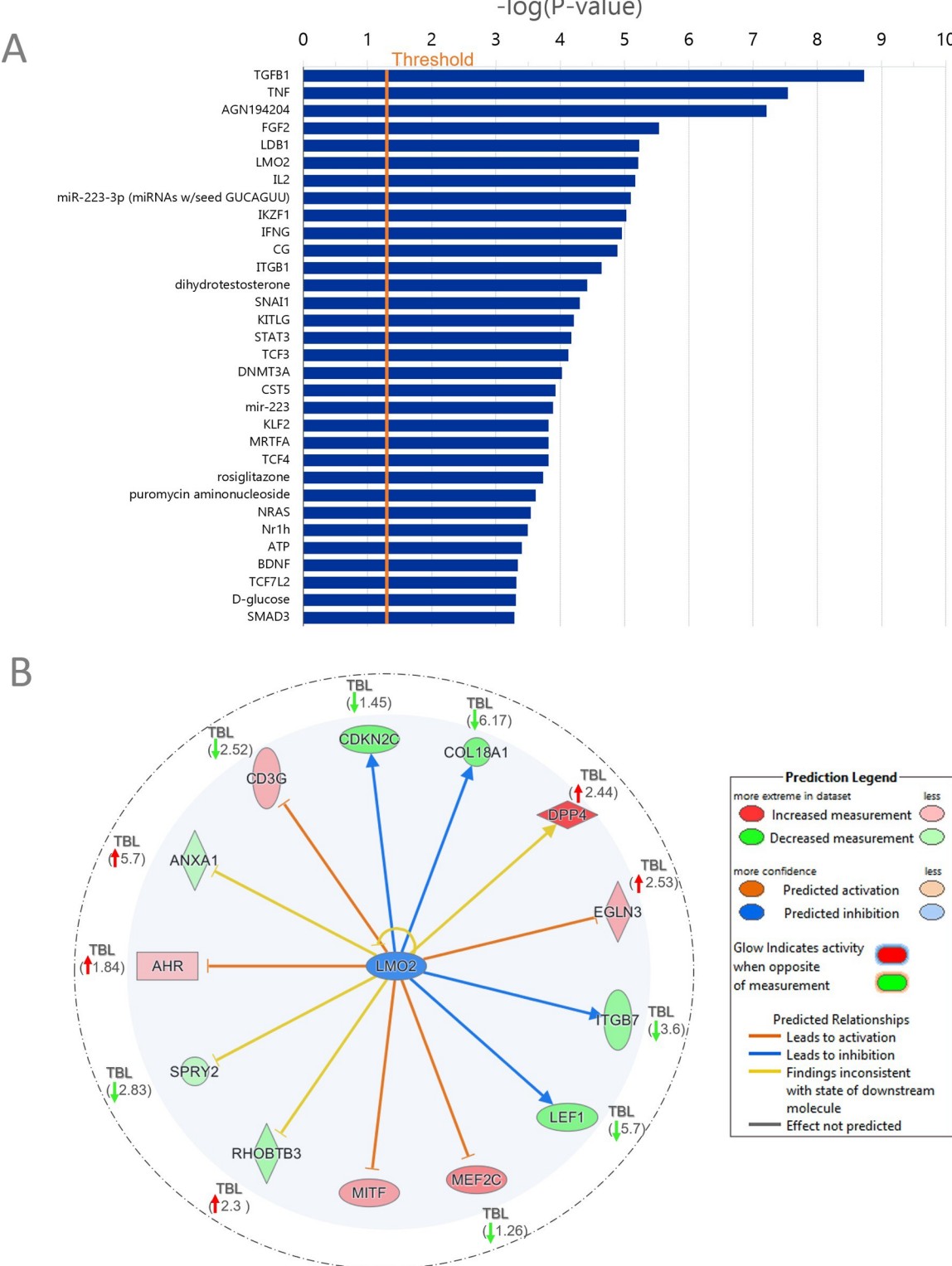

**Fig 7. IPA upstream regulator analysis of TashAT2/IA dataset.** (A) Top upstream regulators identified by IPA. Individual regulators are shown along the y-axis with the significance of the enrichment–log (p-value) shown along the x-axis. Each bar represents the enrichment of genes in a category. The orange vertical line represents the threshold for significance (p = 0.05). (B) LMO2 upstream regulatory network: this network is scored as inhibited in the TashAT2/IA dataset. Molecules denoted as LMO2 targets in the inner shaded circle show relative expression status in TashAT2-BoMac (pKP38): green, reduced; red, elevated. Expression, status and absolute fold change associated with infection for these targets in TBL20 (TBL) are indicated, where identified, within the outer circle.

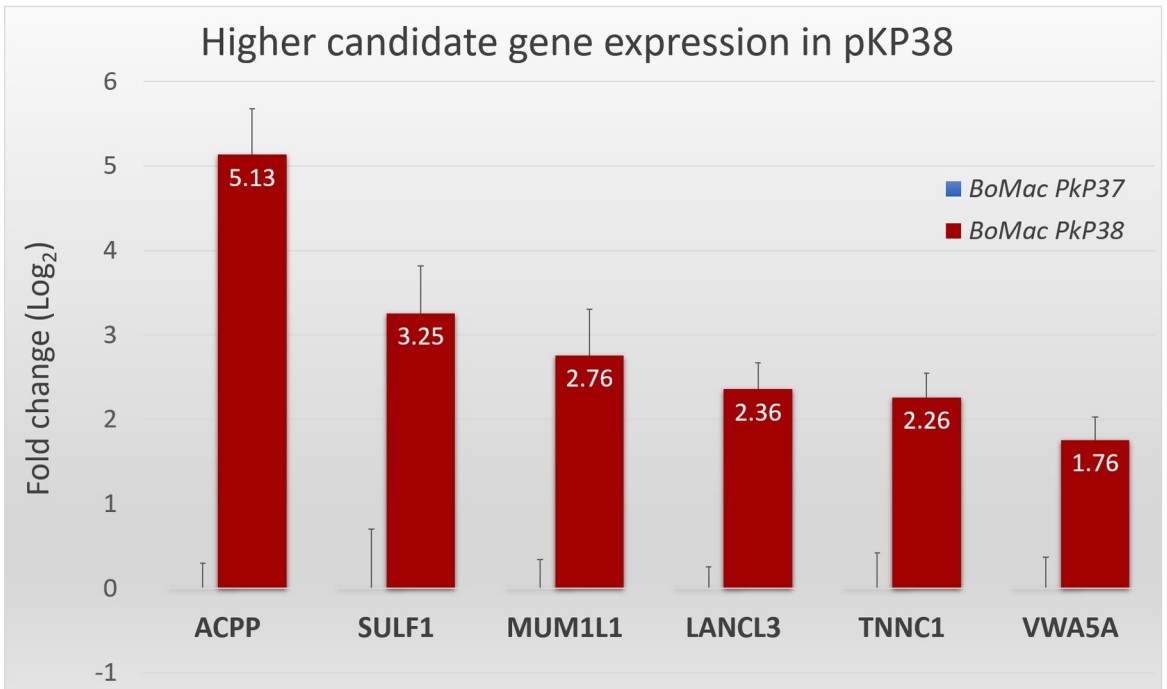

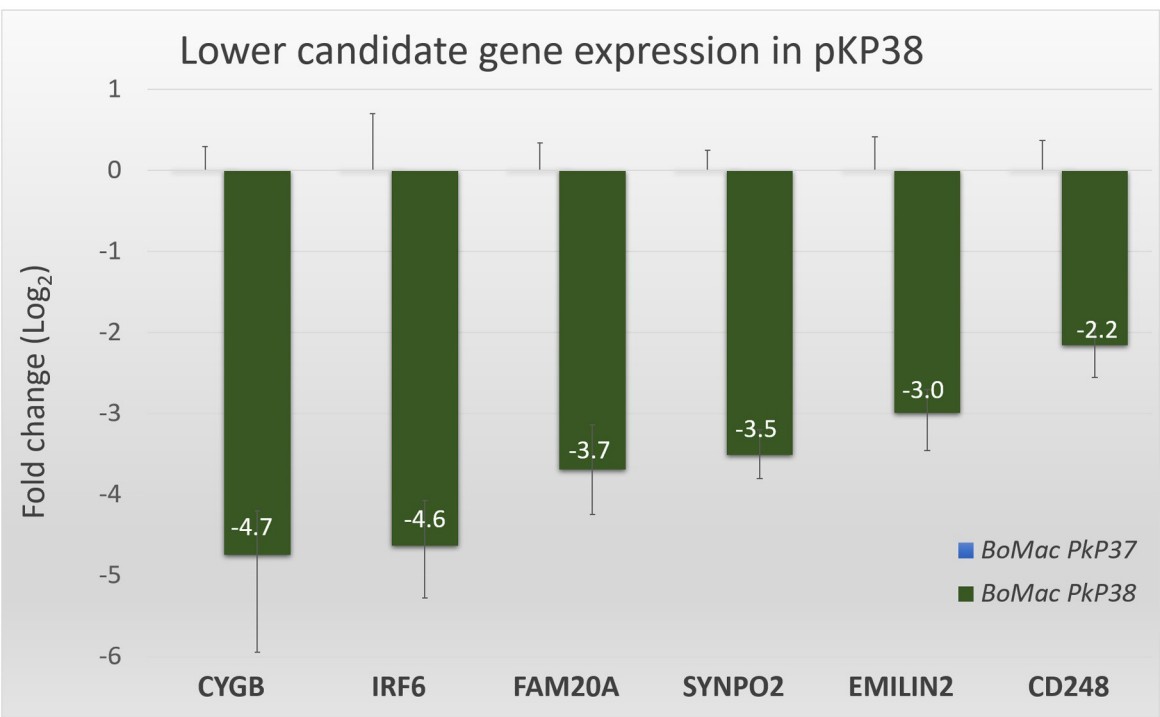

**Fig 8. Real-time qRT-PCR validation of RNA-Seq results.** (A) qRT-PCR of a panel of 6 candidate genes (*ACPP*, *SULF*, *MUM1L1*, *LANCL3*, *TNNC1* and *VWA5A*) displaying higher RNA-Seq levels in BoMac cells expressing TashAT2 (pKP-38) compared to control (pKP37). (B) qRT-PCR of a panel of 6 candidate genes (*CYGB*, *IRF6*, *FAM20A*, *SYNPO2*, *EMILIN2* and *CD248*) displaying lower RNA-Seq values in BoMac cells expressing TashAT2 (pKP-38) compared to control (pKP37). The log fold change in gene expression was calculated via the 2-ΔΔCT method and the error bars represent the standard deviation (n = 3). Expression was normalised using *GAPDH* as a reference gene.

(*VWA5A*) represent infection-associated genes whose expression is up-regulated in TashAT2/BoMac. qRT-PCR confirmed this profile, with statistically significant higher average mRNA levels of 5.13, 2.76 and 1.76 ($\log_2$ fold change), respectively. All three qRT-PCR fold-changes were higher than the fold change estimated by RNA-Seq. Similar results were obtained for non-infection associated TashAT2 modulated candidates, with higher expression confirmed for sulphatase 1 (*SULF1*), Lanthionine synthase C-like protein 3 (*LANCL3*) and troponin (*TNNC1*). Respective $\log_2$ fold change values of 3.25, 2.36 and 2.26 were obtained, which were higher (*SULF1* and *TNNC1*) or comparable to the mean values computed by RNA-Seq (see Fig 8A). For genes with lower expression in the TashAT2/BoMac-DE dataset and parasite infected TBL20, three were selected, namely *FAM20A*, *SYNPO2* and *CD248* (endosialin), a transmembrane glycoprotein linked to inflammation and suppression of proliferation [66, 67]. Validation of lower expression was obtained by qRT-PCR, with $\log_2$ fold changes of -3.7, -3.5 and -2.2, respectively (Fig 8B). A further three non-infection associated genes (*CYGB*, *IRF6* and elastin microfibril interfacer 2 (*EMILIN2*)) were assessed. Lower expression in TashAT2-transfected lines was validated with respective $\log_2$ fold changes of -4.7, -4.6 and -3. Thus, for the majority of genes in the RNA-Seq dataset ($p < 0.05$), the trend of expression indicated is likely to be accurate, although the fold change computed was generally higher for qRT-PCR than the RNA-Seq.

## *GULP1* is a TashAT2 modulated gene associated with differential susceptibility to tropical theileriosis

Previous work by our group has identified a dataset of bovine genes (the H/S-DE dataset) that displayed significant differences in gene expression levels between *T. annulata* infected cell lines derived from *B. taurus* cattle (Holstein) that are susceptible to acute tropical theileriosis relative to lines from *B. indicus* cattle (Sahiwal) that are, in general, tolerant of infection. Moreover, when the genome of *B. taurus* was compared to *B. indicus*, a significant proportion of these genes was found to show differences in the pattern/number of the consensus DNA motif bound by TashAT2 [28]. To investigate this association further, we compared the existing TashAT2/BoMac-DE dataset with the H/S-DE dataset [28]. Of the 837 TashAT2/BoMac-DEGs, 95 overlapped with H/S-DE genes (S9 File) and were, therefore, indicated as TashAT2-modulated and associated with differential susceptibility to tropical theileriosis. The overlap was found to be significant ($p < 0.05$) by Monte Carlo simulation and by calculation of the representation factor: 1.7 ($p = 9.15 \times 10^{-8}$). A number of genes in the overlapping dataset could be linked to pathology generated by parasite infection of hosts of different genomic background. Reactome pathway analysis showed enrichment of genes in "Integrin Signaling" ($p = 0.04$) and Axon Guidance ($p = 0.02$). Genes encoding GULP1, ITGB5 ITGB7, KITLG, LITAF, NTN1 and SLIT2 are all predicted targets of TashAT2 and differentially expressed between Holstein and Sahiwal infected cell lines. In addition, *GULP1*, *ITGB5*, *LITAF*, *NTN1* and *SLIT2* loci were predicted to show different numbers of the TashAT2 consensus binding motifs between *B. indicus* and *B. taurus* genomes [28]. As well as linked to integrin and TGFβ signaling [50, 51], GULP1 has been associated with emphysema [68]. Emphysema is a clinical manifestation of acute theileriosis [10, 69] and differences in a TGFβ autocrine loop have been linked to the increased virulence of Holstein vs Sahiwal infected cells [70]. Moreover, *GULP1* shows the second highest level of elevated expression in TashAT2-BoMac and displays a large difference in the number TashAT2 consensus motifs between the genomes of *Bos indicus* and *Bos taurus* ([28], Fig 9A). We therefore aimed to generate evidence at the protein level for *GULP1* as a target of TashAT2 that displays differential expression between Holstein and Sahiwal infected cells. Immunoblotting with an anti-GULP1 antibody was performed on extracts

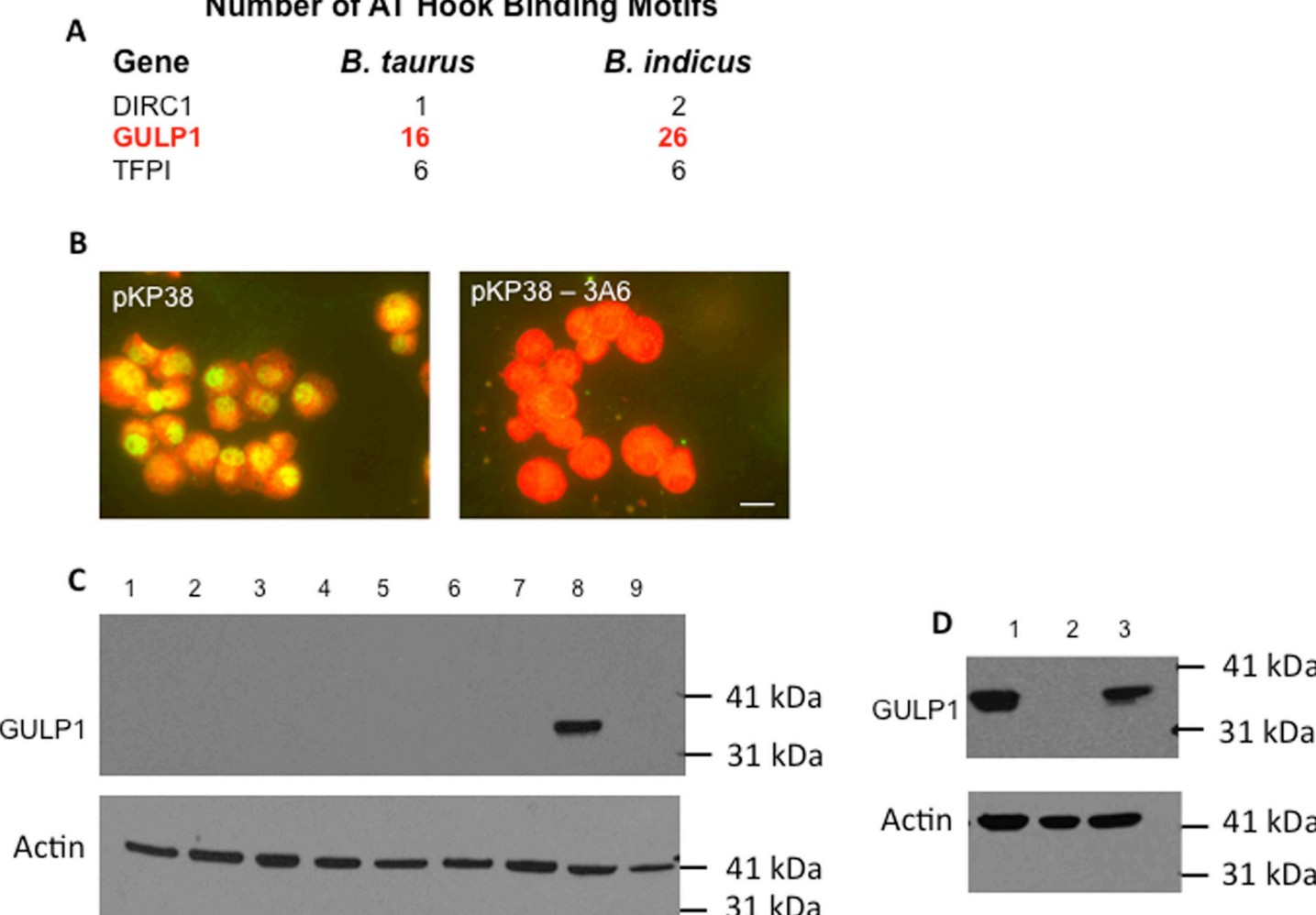

**Fig 9. Reversion of TashAT2 elevated *GULP1* expression in CRISPR-cas9 KO lines and differential expression of *GULP1* in Holstein vs Sahiwal infected cells.** (A) Number of nucleotide motifs predicted to be bound by TashAT2 at the *GULP1* gene locus (red) in *B. taurus* vs *B. indicus* genome, see [28]. (B) IFAT of Anti-TashAT2 against *TashAT2* transfected BoMac (pKP38), and CRISPR-cas9 knock-out (KO) clone (3A6): Bar = 7 μm. (C) Immunoblot with anti-GULP1 and anti-actin (loading control): lane 1–7, extracts of seven pKP38 KO clones; lane 8, pKP38; lane 9, pKP37 (non-expressing control). (D) Immunoblot for GULP1 and actin on extracts of *T. annulata* infected cells from Holstein or Sahiwal cattle: lane 1 Holstein infected cell extract; lane 2, Sahiwal infected cell extract; lane 3, pKP38 extract.

derived from the TashAT2 transfected pKP38 line, the control non-expressing pKP37 line and from 8 cloned lines obtained following CRISPR-Cas9 knock-out of *TashAT2* in pKP38, with lack of TashAT2 validated by IFAT (Fig 9B). A strong band at the size expected for GULP1 was detected in extracts of pKP38 but not pKP37 and, as predicted, knock-out of TashAT2 resulted in a concomitant loss of GULP1 (Fig 9C). Furthermore, immuno-blotting performed on selected Holstein and Sahiwal infected cell line extracts detected GULP1 in the Holstein but not the Sahiwal extract (Fig 9D). We conclude, a marker gene that displays differential expression between infected cells from a breed susceptible or tolerant to theileriosis is a target of TashAT2 modulated expression.

## Discussion

A fundamental requirement for intracellular pathogens is that they must generate a supportive niche within the infected cell. The constructed niche has potential to drive evolution of disease susceptibility. Species of *Theileria* parasite alter their intracellular niche to generate a transformed, proliferating bovine leukocyte. A significant rewiring of the host transcriptome occurs upon leukocyte transformation, but the parasite factors involved have not been conclusively identified. The aim of this study was to robustly investigate the potential of the AT-hook bearing TashAT2 factor to modify the transcriptome of a bovine macrophage derived cell line and assess whether any modifications could be linked to the infected leukocyte and susceptibility to disease. RNA-Seq analysis of three BoMac lines stably expressing the *TashAT2* gene relative to three lines with the gene in the reverse orientation identified 837 bovine genes as differentially expressed. It is unlikely that the large number of significant DE changes across all six lines happened by random chance. Study of the RNA-Seq data indicated that for the majority of genes denoted as significant, differential expression was consistent across all three TashAT2 lines relative to the control lines, indicating a robust calling of DE genes by the algorithm (see S2 File). For all candidate genes tested by qRT-PCR, the profile of DE was validated, albeit indicating that the fold difference from RNA-Seq was almost always lower than that computed by qPCR. This difference in the level of fold change is most likely due to normalisation of transcript abundance by the DESeq2 algorithm, designed to restrict data dispersion, improve sensitivity and precision whilst controlling for false positives [38]. It is unlikely that all TashAT2-DE genes have been captured by the analysis, as stringent criteria for identifying DEGs were used. Hence, the rate of false-negative calling by DESeq2 is necessarily high [71]. This level of stringency may be responsible for the failure to detect the gene encoding the LMO2 transcription factor as differentially expressed, despite strong prediction for reduced LMO2 regulator activity in the analysis of the TashAT2/BoMac-DE dataset (Fig 7) and demonstration of reduced levels of *LMO2* expression in an independent fourth TashAT2 transfected BoMac line [55]. We conclude that for genes indicated as significant in our dataset, the probability of transcriptional modulation is strongly influenced by the presence of the TashAT2 factor.

AT-hook proteins such as HMGA1 and HMGA2 modulate chromatin structure and mediate accessibility of transcriptional regulators to their target genes [72]. However, it is currently unknown if genes within the TashAT2/BoMac-DE set are directly modulated by the binding of TashAT2 to regions proximal to a target gene, i.e. promoters, enhancers or introns. For the mammalian AT-hook protein HMGA2, chromatin immunoprecipitation and sequencing has generated data for modulation of expression by binding to AT rich motifs in the promoters of target gene loci [73]. Other ChIP-Seq studies indicate HMGA proteins bind to AT rich DNA in gene sparse regions of the genome [74]. This does not rule out a role in the control of gene expression, as it has been proposed that AT-hook proteins may function as base composition readers to mould chromosome structure and regulate the epigenome [75]. TashAT2 has an AT-hook arrangement resembling that of HMGA proteins and could operate in a similar manner to alter the probability of gene expression events occurring across the genome. Alternatively, since a number of transcription factors (e.g. TWIST2, SNAI2, LEF1, E2F2 and FOXO3) and the chromatin modulator HDAC9 were identified in our dataset, differential gene expression associated with TashAT2 may be due to secondary transcriptional regulatory events. Whether TashAT2 functions by proximal promoter binding, landscaping the epigenome or co-opting a secondary regulatory network, requires detailed study of its binding sites across the bovine genome.

The results of pathway analysis predicted the most probable phenotype engendered by TashAT2 is modulation of adhesion. Top pathways in both KEGG and IPA were "Focal adhesion", "ECM receptor interaction" and "ILK signaling". Given that most genes identified in the dataset for these pathways primarily show lower relative levels of transcript, the most simplistic prediction is one of reduced functionality for these processes. Pathway analysis also indicated enrichment for genes that regulate the actin cytoskeleton, indicating that TashAT2 could modulate cell shape. These predictions fit well with what is known regarding phenotypic changes of BoMac cells associated with stable expression of TashAT cluster genes. Thus, both TashAT2-BoMac and BoMac expressing the related AT-hook factor, SuAT1, display alteration of cell shape, with less focal processes and a more spread out, flat morphology relative to controls cells [19, 24]. In addition, analysis of the cytoskeletal protein profile in SuAT1-transfected BoMacs indicated reduced levels of alpha actinin, actin and cytokeratin, and increased levels of vimentin [24].

Evidence of a potential role in cell migration/oncogenesis was obtained. As for pathways linked to adhesion, IPA activation status (z-scores) generally predicted that these functions were primarily inhibited in TashAT2 transfected cells. Prediction that TashAT2 generally inhibits oncogenesis can be viewed as contradictory for a role in promoting establishment of the transformed infected leukocyte. There are several possibilities for this apparent dichotomy. Firstly, predictions from our analysis may be influenced by differences in cellular context between TashAT2-transfected BoMac and *Theileria*-infected leukocyte lines. The BoMac line is transformed by expression of the large T antigen of SV40 and *Theileria* macroschizonts do not establish fully in this line. *Theileria* infection induces constitutive activation or suppression of many important host cell transcription regulators [14, 15, 76], and a number of parasite factors with the potential to manipulate the bovine transcriptome of the infected cell have been identified [14, 77–79]. Thus, major differences in the profile of active regulators of gene expression could alter the direction of TashAT2 target gene expression in BoMac relative to infection and skew the results of the pathway predictions.

Secondly, general IPA predictions primarily based on human cancer cell lines may have missed specific mechanisms linked to transformation mediated by *T. annulata*. For example, within the "ILK signaling" pathway it is notable that TashAT2 modulated genes associated with PI3K show elevated expression, and "PI3K-Akt" signaling was enriched in the KEGG pathway analysis. Activation of the PI3K/Akt pathway has been described for *Theileria* infected leukocytes [56] and down-regulation of *SYNPO2* (reduced both by TashAT2 and infection) is reported to promote PI3K/Akt signaling and metastasis [61, 80]. Similarly, a number of genes placed in "Axonal guidance", including several modulated by infection (*SLIT2*, *PLCD1*, SLIT-ROBO Rho GTPase-activating protein 2 (*SRGAP2*) and *TUBA4A*), showed elevated expression. As well as operating in neural development, genes within "Axon guidance" are known to function in cancer and leukocyte chemotaxis/infiltration [81–84].

A third possibility is that the predicted pathways and functions, at least in part, do reflect the role of TashAT2 in the infected cell. A significant overlap between genes modulated by TashAT2 and infection of the host leukocyte was obtained. The top pathways/functions for infection-associated genes modulated by TashAT2 in BoMac included "ILK signaling", "cellular movement", and "reorganisation of cytoskeleton", with the majority of genes in "ILK signaling" and many genes in "cellular movement" showing the same direction of modulation in both datasets. Structural changes to the BL20 cytoskeleton and modulation of cytoskeletal protein levels upon infection have been reported previously, and changes to cell shape associated with parasite infection are obvious in culture [16]. Moreover, a re-cloned infected cell line (D7B12) that displays increased adhesion and altered cell shape relative to the parental D7 clone has reduced expression of TashAT2 [27]. Therefore, TashAT2 may operate to modulate

cell to cell adhesion. How this promotes or enhances infected cell survival is unknown but could aid establishment in the lymph node by altering interaction with uninfected cells of the lymphoid/immune system.

Comparing predictions from our dataset with pathways/molecular functions regulated by vertebrate HMGA proteins generates a considerable degree of overlap. Thus, in a study by Singh *et al*. [73], KEGG pathway analysis of target loci bound by HMGA2 shows striking similarity to our results, with enrichment for genes functioning in cell adhesion and TGFβ1 pathways. Furthermore, like TashAT2, over-expression of HMGA1b alters expression levels of genes encoding integrins and proteins involved in integrin signaling [85], while HMGA1a is known to elevate expression of *KITLG* [86]. Top regulators with a repressed function predicted from the TashAT2/BoMac-DE dataset included TGFβ and SMAD3, while let 7a had one of the highest activation z-scores. These regulators are known to be associated with the function of HMGA proteins in vertebrates. Thus, in epithelial-mesenchymal transition (EMT), which results in altered cell–cell and cell–extracellular (ECM) interactions, TGFβ induces expression of *HMGA2* by SMAD3 mediated inhibition of let 7 micro RNA and in turn HMGA regulates the EMT promoting genes, *TWIST* and *SNAI* (reviewed in [87]). Family members of both *SNAI* and *TWIST* are modulated in TashAT2-BoMac lines (S2 and S4 and S6 Files). Other parallels between pathways/molecular functions modulated by TashAT2, HMGA factors and *Theileria* infection are PI3K/Akt (see above), cell cycle and adipogenesis (including the infection associated *WNT10B* and *HDAC9* genes), HIF1α /Warburg-like glycolysis, development of nervous, muscular and haematological systems and osteoarthritis [87–90]. *Theileria* induces oxidative stress and HIF1α activation that are essential for host leukocyte transformation [88, 89]. Based on these findings we propose that parasite AT-hook factors have the capacity to act as a functional analogue/mimic of mammalian HMGA factors and influence expression of genes that operate in a number of pathways during cell lineage development.

Comparing the TashAT2/BoMac-DE dataset with a dataset of genes linked to disease susceptibility supported a previous postulation that differences between bovine genomes in the pattern of motifs bound by TashAT2 could influence susceptibility to tropical theileriosis. Thus, a subset of genes whose expression is modulated by TashAT2 shows significant expression differences between infected cells derived from either tolerant Sahiwal or susceptible Holstein cattle. Pathway analysis highlighted genes that function in ILK signaling and Axon guidance and included genes that possess different numbers of predicted TashAT2 binding sites between *B. indicus* and *B. taurus*. Further study of *GULP1* expression as an exemplar of a potential susceptibility gene was undertaken. The results confirmed elevated levels of GULP1 protein are strongly linked to expression of TashAT2 in BoMac cells. Moreover, GULP1 protein was clearly detectable in extracts derived from Holstein infected cells but not Sahiwal. Given the known association between GULP1, the TGFβ pathway and pathology previously linked to acute tropical theileriosis, it can be proposed that the marked difference in TashAT2 binding sites at the GULP gene locus may alter susceptibility to acute disease. Moreover, if the AT rich motifs bound by TashAT2 are also recognised by HMGA isoforms, they could influence developmental traits and susceptibility to disease (cancer and immune dysfunction) associated with these mammalian architectural transcription factors [26, 85, 86, 90].

## Conclusions

Intracellular parasites modify the host cell they infect. This study provides strong evidence that genes encoded in the *TashAT* cluster of *T. annulata* act as analogues of mammalian architectural transcription factors to extensively modulate host cell gene expression and generate an intracellular niche that promotes establishment of the transformed leukocyte. The data

supports a previous postulation that differences between host genomes in patterns of DNA motifs bound by TashAT2 generate variability in the epigenome of the infected cell, which in turn influences susceptibility to acute tropical theileriosis. Future work should aim to functionally validate binding sites for TashAT2 across the bovine genome and investigate the role that host-parasite interaction plays in the evolution of differential susceptibility to both infectious and non-infectious disease.

## Supporting information

**S1 Fig. Volcano plot of differentially expressed genes between control and experimental groups.** The significantly differentially expressed genes were represented in red with adjusted P value < 0.05 determined by DESeq2. The x-axis represents $\log_2$change and the y-axis shows–$\log_{10}$ (adjusted p-values).
(DOCX)

**S2 Fig. IPA Integrin-linked kinase (ILK) canonical pathway.** Down-regulated genes are highlighted in green while up-regulated genes are highlighted in red.
(TIFF)

**S1 File. Full list of primers used for real-time quantitative RT-PCR and generation of CRISPR-Cas9 constructs.** The gene symbol, full name, GenBank accession number, forward and reverse primer sequences, length of primers, amplicon length and melting temperature are indicated for each primer pair of individual genes tested. Plasmids and forward and reverse primers for CRISPR-Cas9 are denoted.
(XLSX)

**S2 File. Full list of Differentially Expressed Genes (DEGs).** A Microsoft Excel file containing the full list of significantly DEGs (n = 837, P ≤ 0.05) identified by DESeq2 analysis. It contains: Ensembl stable gene ID, Gene name, Gene description, $\log_2$ FC and p-value. Red highlight indicates raw expression data where values are lower across all three PKP38 samples relative to PKP37; Green highlight indicates where values are consistently higher in PKP38 relative to PKP37.
(XLSX)

**S3 File. Full list of KEGG pathways identified in TashAT2 modulated DEGs list.**
(XLSX)

**S4 File. Top 100 canonical pathways identified by Ingenuity Pathway Analysis (IPA) in TashAT2 modulated DEGs list (p ≤ 0.05).** The file contains Canonical Pathway, -log(p-value) Ratio, z-score, Molecules identified within the pathway.
(XLSX)

**S5 File. Full list of biological functions and diseases categories identified by Ingenuity Pathway Analysis (IPA).** The file contains Category, Diseases or Functions Annotation, P-values for overrepresentation, predicted Activation State, Activation z-score, a list of Molecules within the category.
(XLSX)

**S6 File. Full list of upstream regulators and regulatory networks identified by IPA in TashAT2 modulated DEGs list (p ≤ 0.05).**
(XLSX)

**S7 File. Microsoft Excel file containing full list of infection associated genes (n = 209) overlapping with TashAT2 modulated genes dataset.** The direction of modulation of gene

expression is denoted by coloured arrows: green = down, red = elevated.
(XLSX)

**S8 File. A Microsoft Excel file of infection associated overlapping genes list (n = 209) showing top 50 canonical pathways and biological function or diseases categories.**
(XLSX)

**S9 File. Microsoft Excel file containing full list of 95 genes (p < 0.05) overlapping between TashAT2/BoMac-DE and the H/S-DE dataset of Larcombe *et al*., 2022 [28].** Red denotes elevated expression in pKP38 (TashAT2) line; green denotes reduced expression in pKP38.
(XLSX)

## Author Contributions

**Conceptualization:** Zeeshan Durrani, Brian Shiels.

**Data curation:** Zeeshan Durrani, Chew Weng Cheng.

**Formal analysis:** Zeeshan Durrani, Jane Kinnaird, Chew Weng Cheng, Francis Brühlmann, Andrew Jackson, Stephen Larcombe, William Weir.

**Funding acquisition:** Brian Shiels.

**Investigation:** Zeeshan Durrani, Jane Kinnaird, Paul Capewell, Stephen Larcombe, Brian Shiels.

**Methodology:** Zeeshan Durrani, Chew Weng Cheng, Francis Brühlmann, Paul Capewell, Andrew Jackson, Philipp Olias.

**Project administration:** Brian Shiels.

**Validation:** Jane Kinnaird, Brian Shiels.

**Visualization:** Brian Shiels.

**Writing – original draft:** Zeeshan Durrani, Brian Shiels.

**Writing – review & editing:** Zeeshan Durrani, Jane Kinnaird, Chew Weng Cheng, Francis Brühlmann, Paul Capewell, Andrew Jackson, Stephen Larcombe, Philipp Olias, William Weir, Brian Shiels.

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
