## [Decision Letter · Decision Letter 0]

27 Mar 2023

PONE-D-23-06811A parasite DNA binding protein with potential to influence disease susceptibility acts as an analogue of mammalian HMGA transcription factorsPLOS ONE

Dear Dr. Shiels,

Thank you for submitting your manuscript to PLOS ONE. After careful consideration, we feel that it has merit but does not fully meet PLOS ONE’s publication criteria as it currently stands. Therefore, we invite you to submit a revised version of the manuscript that addresses the points raised during the review process.

We look forward to receiving your revised manuscript.

Kind regards,

Muhammad Mazhar Ayaz, Ph.D

Academic Editor

PLOS ONE

Journal Requirements:

Reviewers' comments:

Reviewer's Responses to Questions

**Comments to the Author**

1. Is the manuscript technically sound, and do the data support the conclusions?

Reviewer #1: Yes

2. Has the statistical analysis been performed appropriately and rigorously? 

Reviewer #1: Yes

3. Have the authors made all data underlying the findings in their manuscript fully available?

Reviewer #1: Yes

4. Is the manuscript presented in an intelligible fashion and written in standard English?

Reviewer #1: Yes

5. Review Comments to the Author

Reviewer #1: Abstract

Introductory part (line 24-34) is too long. It can be cut short and can be discussed in detail in the introduction or discussion sections.

it is suggested to use past tense while writing the abstract and result description.

lines 39-41... "Validation at the protein level of modulated expression of a gene (GULP1) potentially linked to disease susceptibility was obtained for TashAT2 transfected BoMac, and between an infected Sahiwal and Holstein cell line".... what were the findings???/ rather than describing the methadology, mention the major findings.

Introduction is too long. Restrict it to the back ground of study and take explanations to discussion.

Methods

please provide the country along with the suppliers for cell lines and chemicals.

Please provide the web links for the databases used in this study.

Results

Table 1 and 2 are cropped and can not be read completely.

6. PLOS authors have the option to publish the peer review history of their article (what does this mean?). If published, this will include your full peer review and any attached files.

Reviewer #1: No

---

## [Author Response · Author response to Decision Letter 0]

19 Apr 2023

Response to Reviewers comments

Reviewer #1:

Abstract

Introductory part (line 24-34) is too long. It can be cut short and can be discussed in detail in the introduction or discussion sections.it is suggested to use past tense while writing the abstract and result description.

Response: The abstract has been shortened and the past tense has now been used to describe results here and throughout the manuscript: see lines 24-41.

lines 39-41... "Validation at the protein level of modulated expression of a gene (GULP1) potentially linked to disease susceptibility was obtained for TashAT2 transfected BoMac, and between an infected Sahiwal and Holstein cell line".... what were the findings???/ rather than describing the methadology, mention the major findings.

Response: This sentence has been rewritten to describe the findings: lines 37-39 “Altered protein levels encoded by one of these genes (GULP1) was strongly linked to expression of TashAT2 in BoMac cells and was demonstrated to be higher in infected Holstein leucocytes compared to Sahiwal.”

Introduction is too long. Restrict it to the back ground of study and take explanations to discussion.

Response: The introduction has been substantially shortened (by 17 lines), and the explanations made moved/merged with end of discussion: lines 698-699 and lines 709-712 (revised document).

Methods

please provide the country along with the suppliers for cell lines and chemicals.

Please provide the web links for the databases used in this study.

Response: A reference to the generation of the infected cell lines was added to methods section, and the country of origin of the T. annulata isolate used to generate the lines stipulated: see lines 101-103 “The infected bovine cell lines derived from Sahiwal (S3) and Holstein (H3) cattle were generated and cultured as described [28, 43], and represent the Hissar (India) isolate of T. annulata.”

Suppliers and country for chemicals/reagents/kits have been added to the methods section, throughout. References, web links and accession numbers have been added to the data availability section: lines 257-263.

Results

Table 1 and 2 are cropped and can not be read completely.

Response: The issue with Tables was due to their placement in the body text and they were cropped when the journal website built the PDF; the Tables have been reformatted and can be seen fully in the tracked/corrected Word document version of the manuscript. If further modification is required or they need to be loaded up as separate files to be observed, we will be happy to do so.

---

## [Editor Report · Decision Letter 1]

18 May 2023

A parasite DNA binding protein with potential to influence disease susceptibility acts as an analogue of mammalian HMGA transcription factors

PONE-D-23-06811R1

Dear Dr. Shiels,

We’re pleased to inform you that your manuscript has been judged scientifically suitable for publication and will be formally accepted for publication once it meets all outstanding technical requirements.

Kind regards,

Muhammad Mazhar Ayaz, Ph.D

Academic Editor

PLOS ONE
---

## [Editor Report · Acceptance letter]

26 May 2023

PONE-D-23-06811R1 

A parasite DNA binding protein with potential to influence disease susceptibility acts as an analogue of mammalian HMGA transcription factors 

Dear Dr. Shiels:

I'm pleased to inform you that your manuscript has been deemed suitable for publication in PLOS ONE. Congratulations! Your manuscript is now with our production department. 

Kind regards, 

on behalf of

Dr. Muhammad Mazhar Ayaz 

Academic Editor

PLOS ONE